# SELU: Self-Learning Embodied Multimodal Large Language Models in Unknown Environments

**Boyu Li**[1,2,3], **Haobin Jiang**[4], **Ziluo Ding**[3], **Xinrun Xu**[5], **Haoran Li**[1,2],
**Dongbin Zhao**[1,2], **Zongqing Lu**[4*]

[1] *Institute of Automation, Chinese Academy of Sciences*

[2] *School of Artificial Intelligence, University of Chinese Academy of Sciences*

[3] *Beijing Academy of Artificial Intelligence*

[4] *School of Computer Science, Peking University*

[5] *Institute of Software, Chinese Academy of Sciences*

**Reviewed on OpenReview:** *https://openreview.net/forum?id=G5gROx8AVi*

## Abstract

Recently, multimodal large language models (MLLMs) have demonstrated strong visual understanding and decision-making capabilities, enabling the exploration of autonomously improving MLLMs in unknown environments. However, external feedback like human or environmental feedback is not always available. To address this challenge, existing methods primarily focus on enhancing the decision-making capabilities of MLLMs through voting and scoring mechanisms, while little effort has been paid to improving the environmental comprehension of MLLMs in unknown environments. To fully unleash the self-learning potential of MLLMs, we propose a novel actor-critic self-learning paradigm, dubbed SELU, inspired by the actor-critic paradigm in reinforcement learning. The critic employs self-asking and hindsight relabeling to extract knowledge from interaction trajectories collected by the actor, thereby augmenting its environmental comprehension. Simultaneously, the actor is improved by the self-feedback provided by the critic, enhancing its decision-making. We evaluate our method in the AI2-THOR and VirtualHome environments, and SELU achieves critic improvements of approximately 28% and 30%, and actor improvements of about 20% and 24% via self-learning.

## 1 Introduction

Multimodal Large Language Models (MLLMs) have demonstrated impressive perceptual and understanding capabilities across various domains, *e.g.,* , web applications (Ma et al., 2024; Tao et al., 2024; Liu et al., 2024), robotics (Xiong et al., 2024; Li et al., 2024c), gaming (Li et al., 2024d; Qi et al., 2024; Xu et al., 2024), and autonomous driving (Wen et al., 2024; Zhang et al., 2024). Thanks to their powerful capabilities, many works, *e.g.,* , Jarvis-1 (Wang et al., 2023b), STEVE-1 (Lifshitz et al., 2023), and Cradle (Tan et al., 2024b), directly utilize the pre-trained MLLMs to complete various decision-making tasks in different embodied environments.

However, the generalization ability of existing pre-trained MLLMs cannot meet the needs of all environments. For some uncommon environments, embodied MLLMs often exhibit hallucinations and poor visual understanding (Huang et al., 2024; Jiang et al., 2024a). For example, they cannot distinguish left from right or fail to recognize where objects are (Tan et al., 2024b). The reason is that MLLMs have not been further grounded with the environments (Su et al., 2022; Sun et al., 2024b). Grounding can be realized by fine-tuning on the experiences from interacting with the environments. Based on the evaluation methods, experience can be categorized into three types: human feedback (Dai et al., 2024; Kirk et al., 2024), environmental feedback (Tan et al., 2024a; Wang et al., 2024b), and self-feedback (Pang et al., 2024; Madaan et al.,

---

*Correspondence to Zongqing Lu <zongqing.lu@pku.edu.cn>.

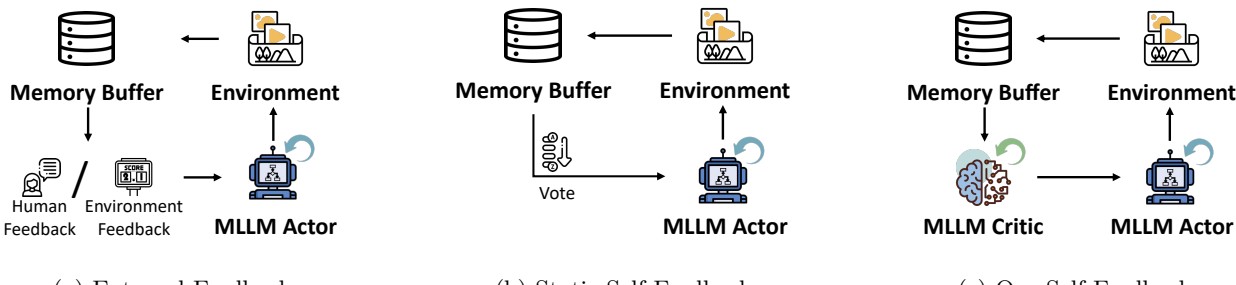

(a) External Feedback        (b) Static Self-Feedback        (c) Our Self-Feedback

Figure 1: Comparison of our framework with other frameworks in terms of the feedback type.

2023). The first two types require additional efforts as illustrated in Figure 1a. Human feedback requires expert demonstrations and annotations, which can be costly and influenced by individual biases (McAleese et al., 2024). Environmental feedback assumes we can obtain a dynamics model of the environment or a well-designed reward model (Fan et al., 2022; Urcelay et al., 2024; Jiang et al., 2024b). Unfortunately, many environments including the real world do not meet such requirements, so these grounding methods are not general. Therefore, we are committed to finding a *general way* to fill in the remaining gaps.

When operating in an unknown embodied environment without external feedback, the MLLM must rely solely on its inherent capabilities. An unknown environment refers to one in which the MLLM has neither been exposed to its visual data during pretraining, nor fine-tuned on its task-specific data to achieve grounding. Some work, see Figure 1b, utilizes the evaluation (discriminative) ability of the pre-trained model itself (Pang et al., 2024) or statistics like voting (Wang et al., 2023a; Huang et al., 2023) to evaluate its own decision-making, and uses such self-feedback to enhance the model's decision-making (generative) capabilities. However, it is evident that this form of self-learning is limited by the model's **static** evaluation capacity, restricting its ability to enhance decision-making. Ideally, we hope that self-learning can work similarly to the actor-critic paradigm (Konda & Tsitsiklis, 1999) in reinforcement learning for embodied tasks, where both the actor and critic iteratively refine their performance. If so, the potential for actor enhancement can be greatly expanded. Unlike reinforcement learning, which relies on external rewards for training, we do not assume the access to any external feedback. Therefore, we aim to **develop a new actor-critic self-learning paradigm for embodied MLLMs in unknown environments.**

In this paper, we introduce a novel **SE**lf-**L**earning paradigm in **U**nknown environments, dubbed **SELU**, as illustrated in Figure 1c. Inspired by the actor-critic paradigm in reinforcement learning, our paradigm learns to simultaneously optimize the MLLM's ability to understand the environment and to make decisions. For the actor module, we fine-tune the model based on the self-feedback from the critic. As the actor gets improved, it can roll-out more successful trajectories to fine-tune the critic. However, without environmental feedback, the critic may provide inaccurate feedback at the beginning of the training phase, which might mislead the overall optimization. Therefore, we adopt self-asking to correct self-feedback and leverage hindsight relabeling to increase sample efficiency by turning the failure trajectory into a successful one for other tasks. These high-quality and diverse trajectories are deemed to enhance the critic's comprehension of the environment. Ultimately, the coupling of these two components mutually promotes the improvement of each other, unleashing the full self-learning potential of MLLMs.

Our key contributions can be summarized as follows:

- We propose SELU, a self-learning paradigm for embodied MLLMs, inspired by the actor-critic paradigm in reinforcement learning, which enables MLLMs to improve themselves in unknown environments without external feedbacks.

- We leverage self-asking and hindsight relabeling to achieve the improvement of the critic, which greatly increase the sample efficiency of our algorithm and make the self-learning possible.

- We demonstrate the effectiveness of SELU in AI2-THOR and VirtualHome, achieving critic improvements of approximately 28% and 30%, and actor improvements of about 20% and 24%, respectively.

## 2 Related Work

### 2.1 MLLMs with External Feedback

In recent years, MLLMs has achieved impressive results across various visual benchmarks (Mathew et al., 2021; Tang et al., 2024; Lu et al., 2024b), demonstrating remarkable perception and decision-making capabilities. However, these models still exhibit flaws and often generate unexpected outputs, such as perceptual hallucinations and unreasonable decisions (Yu et al., 2024; Chen et al., 2024b). Inspired by reinforcement learning, current approaches use external feedback to correct MLLM's erroneous outputs through a cycle of interaction, feedback, and correction (Pan et al., 2024; Gero et al., 2023). RL methods such as PPO (Schulman et al., 2017) and GRPO (Shao et al., 2024) have proven effective (Sun et al., 2024a; Zhai et al., 2024; Lu et al., 2024a). Generally, there are two sources of external feedback: human preference feedback and environmental feedback. Human preference data is used to train a reward model aligned with human preferences (Ouyang et al., 2022). In contrast, environmental feedback typically stems from a rule-based reward function (Tan et al., 2024a; Guo et al., 2025) or a pre-trained large model (Rocamonde et al., 2023; Lee et al., 2024), which often requires substantial support from expert data. For instance, Konyushkova et al. (2025) help Gemini 1.5 to understand Dialog Games through a rule-based approach, generate a positive evaluation function, and construct a high-quality dataset to improve its image understanding capabilities. Gaven et al. (2024) uses the SAC (Gaven et al., 2024) and a sparse context reward from the environment to improve the plan ability of LLMs in text-based household environments. Compared to existing studies, we focus on the self-learning potential of MLLMs, as external feedback may be not professional or available in unknown environments.

### 2.2 Self-Improvement in LLMs

Self-improvement in Large Language Models (LLMs) has gained significant attention, as researchers strive to develop models that can learn and adapt from their own outputs, interactions, and internal feedback mechanisms, without relying on external human-labeled data (Yan et al., 2023; Haluptzok et al., 2023). Early explorations in this area are based on unsupervised learning techniques, where models learn representations from vast datasets without explicit human guidance (Winter et al., 2022; Zhao et al., 2019). Expanded to LLMs, self-improvement goes further by enabling models to critique, refine, and adapt their behavior in a more autonomous manner (Tan et al., 2023; Choi et al., 2024). There are two common self-improvement methods: prompt engineering and fine-tuning. The former is an efficient and intuitive approach for large-scale LLMs, as it allows for the establishment of various chains of thought (CoTs) (Wei et al., 2022) to address the same problem (Huang et al., 2023; Feng et al., 2023). For instance, Madaan et al. (2023) demonstrate that LLMs can enhance the rationality of responses by simultaneously inputting a question and reflecting on previous answers. The latter one is proposed as the prompt engineering is unstable for small-scale LLMs, and shows limitation to solve long-horizon tasks (Huang et al., 2023; Pang et al., 2024; Wang et al., 2024a). Wang et al. (2024a) develop a fine-tuning dataset by generating synthetic negative responses to optimize the LLM evaluator, while other work explores how to improve the instruction-following abilities of LLMs (Huang et al., 2023; Pang et al., 2024). Based on these studies, we choose to use fine-tuning to optimize small-scale MLLMs. However, existing methods focus on natural language tasks and overlook the enhancement of MLLM's understanding capabilities in embodied environments. Therefore, we employ an actor-critic framework to facilitate comprehensive self-learning in MLLMs, optimizing both perception and decision-making abilities.

## 3 Preliminaries

### 3.1 Actor-Critic in Reinforcement Learning

Actor-critic (Konda & Tsitsiklis, 1999) is a widely adopted framework in reinforcement learning (Li et al., 2024a). The agent consists of two learning modules: an actor and a critic, which are optimized iteratively. The actor selects and executes actions based on current observations. The critic evaluates these observations (and actions) by estimating their values based on reward signals received from the environment, thereby

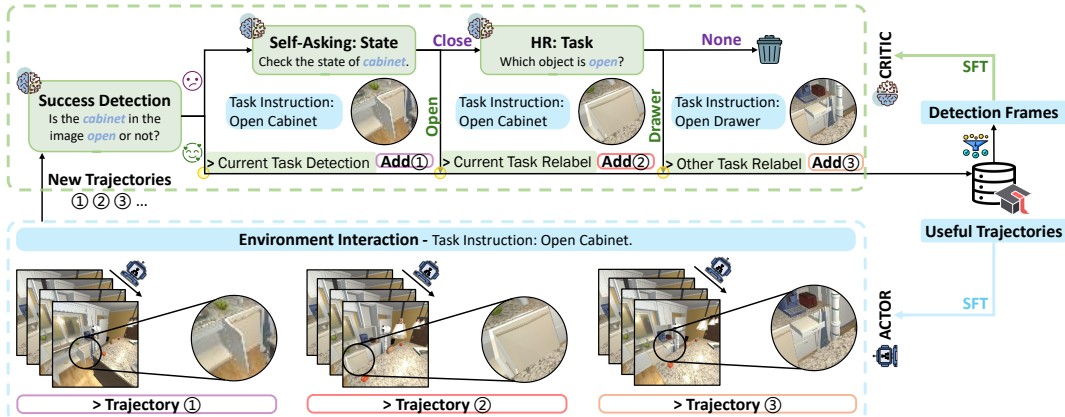

Figure 2: The framework of SELU. (*lower*) The actor MLLM, represented as a robot, collects trajectories for the given instructions. (*upper*) The critic MLLM, denoted as a brain, evaluates these trajectories and determines whether they complete the tasks, guiding the update of the actor MLLM. In addition, the critic MLLM implements self-asking and hindsight relabeling to build a dataset for optimizing itself. The whole framework does not require any external feedback, such as environmental rewards or human annotations.

guiding the actor to make improved choices in future. This framework takes advantage of both policy-based learning and value-based learning and is popular nowadays like SAC (Haarnoja et al., 2018).

## 3.2 Actor-Critic for MLLMs

With the development of MLLMs, the feedback provided to the agent is no longer constrained to scalar values, like rewards; it can now include diverse modalities, such as natural language (Dong et al., 2024). This enables the critic gain more specific and informative feedback on the outputs of the actor. Consequently, it can provide more accurate guidance for actor improvement. A prevalent approach in this domain is incorporating human feedback into the critic and building a static evaluation module that can reflect human preference (Ouyang et al., 2022; Kirk et al., 2024). Specifically, human annotated data is used to train a critic model (McAleese et al., 2024) or a reward model (Sontakke et al., 2023; Wang et al., 2024b) to align the MLLM with human preferences better. More rigorous approaches leverage external evaluation mechanisms, such as tool-interactive learning (Gou et al., 2024; Chen et al., 2021), or external knowledge sources like Wikipedia and the Internet (Xu et al., 2023; Li et al., 2024b).

However, regardless of whether preference labels or external tools are used, human intervention remains inevitable. To overcome this reliance, methods like self-consistency (Wang et al., 2023a; Schick et al., 2023) employ a voting mechanism to enable the model to evaluate its own behavior without relying on external information. In addition, LLM-as-a-Judge approaches utilize an LLM as a evaluator to eliminate human intervention (Bai et al., 2022; Lee et al., 2023). However, both self-consistency and LLM-as-a-Judge lack a learnable critic module and therefore cannot improve the model's grounding knowledge of the environment it interacts with. In contrast to previous work, we propose a novel actor-critic based paradigm aimed at achieving self-learning for both the actor MLLM and critic MLLM, enabling them to iteratively improve decision-making and grounding abilities without external human feedback or environmental rewards.

## 4 Method

The framework of our method is shown in Figure 2. It consists of two components: the actor MLLM and the critic MLLM. The actor MLLM follows instructions and collects trajectories in the environment. The critic MLLM evaluates the collected trajectories and acquires bootstrapped data via self-asking and hindsight relabeling to optimize itself (Section 4.1). Guided by the success detection results from the critic MLLM, the actor MLLM subsequently improves its decision-making performance in the environment (Section 4.2).

By combining the two processes, we can achieve coupled improvements of the critic MLLM and the actor MLLM (Section 4.3) without reliance on external feedback.

## 4.1 Critic: Self-Asking and Hindsight Relabeling

As introduced in Section 1, enhancing the interpretation of environmental grounding information is crucial for an MLLM to improve its performance. In our framework, we propose to achieve this objective via self-asking and hindsight relabeling to acquire bootstrapped data for optimizing the critic MLLM.

Specifically, given an instruction $I$, the actor MLLM collects a trajectory by following this instruction. The critic takes the last frame $o_T$ of this trajectory as input, using it as the detection frame to determine whether the task depicted by $I$ is completed,

$$l_d = M_c(I, p_d, o_T), \tag{1}$$

where $M_c$ denotes the critic MLLM, $l_d \in \{\text{"}yes\text{"}, \text{"}no\text{"}\}$ is the result of the success detection, and $p_d$ is a prompt for the detection. If the detection result is $l_d = \text{"}yes\text{"}$, it means this trajectory is considered to be a successful sample by the critic MLLM for the given instruction and store this trajectory directly into the critic fine-tuning dataset $\mathcal{D}_{\text{critic}}$ in the format $(I, p_d, o_T, l_d)$, as shown by trajectory 1 in Figure 2. This trajectory includes environmental grounding information that aligns with the knowledge contained in the critic MLLM.

If the detection result is $l_d = \text{"}no\text{"}$, it means that the critic MLLM considers the trajectory unsuccessful in fulfilling instruction $I$. We first apply **self-asking** to examine the state of task-related objects, as the decision made by the critic MLLM might not be precise due to potential hallucinations. The critic MLLM is used to obtain the object states,

$$l_d' = M_c(l_s, I), \quad l_s = M_c(j_I, p_s, o_T) \tag{2}$$

where $j_I$ is the object name extracted from instruction $I$ by a text-processing function, $p_s$ is the prompt for object state analysis, $l_s$ is the analysis result, and $M_c$ provides a new success detection $l_d'$ based on $l_s$ and $I$. The format is corrected to $(I, p_d, o_T, l_d')$ and stored into the critic fine-tuning dataset $\mathcal{D}_{\text{critic}}$ if $l_d' = \text{"}yes\text{"}$. For example, in trajectory 2 of Figure 2, the critic MLLM initially misjudged the completion of the "open cabinet" task. However, when prompted to focus on the state of the cabinet, it successfully self-corrected its judgment.

If the critic MLLM still considers the trajectory as failure, we propose to use **hindsight relabeling** to make use of this trajectory, since it might be helpful for learning the environmental grounding of other instructions. Hindsight relabeling is a method that originated in goal-conditioned reinforcement learning (Andrychowicz et al., 2017). It is based on a simple principle: if a trajectory does not complete the target task, it can be viewed as having accomplished other tasks or subtasks. For example, as shown in trajectory 3 in Figure 2, although it does not complete the task "open cabinet", it successfully completes another task "open drawer". Therefore, we relabel this trajectory with the instruction "open drawer" to help the critic MLLM recognize the completion of the relabeled instruction. We can write this process as,

$$I' = M_c(l_h, a_I), \quad l_h = M_c(a_I, p_h, o_T) \tag{3}$$

where $a_I$ is the verb extracted from the instruction $I$ by another text-processing function. All text-processing functions rely on a classic format-matching approach, requiring the MLLM to strictly follow the prompt format in its output. $p_h$ is the prompt for hindsight relabeling, and $l_h$ is the output, which is usually an object name or None. $M_c$ checks whether any objects, other than the target object, in the observation $o_T$ have completed the task associated with $a_I$. $M_c$ generates a new instruction $I'$ if $l_h$ is not None. This enables the generation of new instructions, with "closed-book" actions restricted to the robot's capabilities, and "open-book" objects identified independently by the MLLM itself. After that, the data $(I', p_d, o_T, \text{"}yes\text{"})$ is stored into the critic fine-tuning dataset $\mathcal{D}_{\text{critic}}$. Finally, if a failed trajectory proves meaningless after hindsight relabeling, it is considered as not helpful for the MLLM to understand the environment and discarded.

By applying self-asking and hindsight relabeling, a critic fine-tuning dataset $\mathcal{D}_{\text{critic}}$ is created by MLLM itself containing the last frames considered as successful and the last frames relabeled as successful after self-asking and hindsight relabeling.

## 4.2 Actor: Critic-Guided Improvement

Recent work has shown that the discriminative ability of an LLM exceeds its generative ability (Pang et al., 2024). As MLLMs are typically trained the same way as LLMs, we believe that MLLMs' evaluation abilities would also surpass their generation abilities. For instance, we can easily prompt MLLMs to extract understanding from a given image, but it is challenging to prompt them to choose an appropriate action based on perceived task-relevant information like distance or direction. Our experiment result in Section 5.2 also supports this conclusion, where the critic module always performs better than the actor. Therefore, we propose using the critic MLLM to guide the improvement of the actor MLLM in the environment without external feedback.

Specifically, the actor MLLM interacts with the environment and collects online trajectories. At each timestep $t$, the actor generates an action plan $l_{a,t}$ by,

$$l_{a,t} = M_a(I, p_a, o_t), \tag{4}$$

where $M_a$ represents the actor MLLM, $I$ is the task instruction, $p_a$ is the prompt for action plan and $o_t$ is the current image observation. After collecting a whole trajectory, the critic MLLM determines whether this trajectory completes the instruction $I$, as described in Section 4.1. If the answer is yes, the whole trajectory is put into the actor fine-tuning dataset $\mathcal{D}_{\text{actor}}$ with a format of $\{(I, p_a, o_t, l_{a,t})\}_{t=0}^T$. The relabeled successful trajectories after hindsight relabeling are also added into the actor fine-tuning dataset $\mathcal{D}_{\text{actor}}$. Since this dataset only contains task completion trajectories, the actor can quickly converge towards completing tasks in the current environment by fine-tuning on the dataset. Note that the actor fine-tuning dataset $\mathcal{D}_{\text{actor}}$ consists of trajectory data, while the critic fine-tuning dataset $\mathcal{D}_{\text{critic}}$ only contains the last frames of these trajectories.

## 4.3 Actor-Critic Coupling Improvement

We employ Supervised Fine-Tuning (SFT) (Devlin et al., 2019; Brown et al., 2020) and Low-Rank Adaptation (LoRA) (Hu et al., 2022) to update both the actor and critic MLLMs. Initially, the actor MLLM interacts with the environment to collect online trajectories containing grounding information, as depicted in the lower part of Figure 2. The critic module then evaluates and classifies these trajectories based on the last frame, as shown in the upper part of Figure 2. We select successful trajectories identified by the critic MLLM to create the actor fine-tuning dataset $\mathcal{D}_{\text{actor}}$. Subsequently, we utilize the last frame to construct the critic fine-tuning dataset $\mathcal{D}_{\text{critic}}$. The update of the actor and critic can be performed iteratively, and make them both improve step-by-step. A detailed pseudo-code for our algorithm is available in Appendix A.1.

# 5 Experiments

## 5.1 Experimental Setup

**Environments.** In order to simulate embodied MLLM interactions in unknown environments, we select AI2-THOR (Kolve et al., 2022) and VirtualHome (Puig et al., 2018) for our experiments. Both environments offer open-ended tasks, various interactive objects, and selectable camera perspectives, facilitating data collection for the actor and critic. More details can be seen in Appendix A.2.1.

- **AI2-THOR** is an interactive simulation environment designed for embodied AI research. The primary tasks require agents to navigate and interact with household objects. It offers highly realistic 3D environments that simulate kitchens, living rooms, and other indoor settings.

- **VirtualHome** is also an embodied simulation platform designed to imitate human activities and tasks in home environments. This environment focuses on task completion through multi-step action sequences, making it ideal for testing long-term planning.

**Task Selection.** The goal of SELU is to enable the MLLM to acquire atomic skills in unknown environments through self-feedback. Therefore, the instruction list is derived from the robot's executable actions. In AI2-

THOR, locobot is selected as the agent, as this work does not consider low-level control of robotic arms. There are three fundamental actions: pick up, open, and break, which serve as the basis for long-horizon task composition. To ensure task diversity and feasibility, we first prompt the MLLM to explore the environment and use the discovered objects to initialize the instruction list which serves as the task set. We randomly sample 2-3 objects for each type of task. Considering the requirements of atomic skills and training costs, we restrict the maximum step for all tasks to 10. We apply a similar approach in the VirtualHome environment, selecting "female1" as the agent and primarily testing in grab, open, and sit tasks.

**MLLMs.** To demonstrate the generalization capability of our framework, we conduct experiments using two MLLMs: LLaVA (Liu et al., 2023) and Qwen-VL (Bai et al., 2023).

- **LLaVA** has gained prominence as one of the most popular MLLMs due to its simple architecture and lower training data requirements. These features enable LLaVA to generate responses more swiftly, and suitable for inference to investigate self-learning.

- **Qwen-VL** is the first model to use a 448x448 resolution image input. Due to its higher resolution, this model exhibits enhanced visual understanding capabilities. We opt for Qwen-VL with the aim of better success detection, thereby facilitating more efficient self-learning of MLLMs.

**Baselines.** We compare SELU with five methods to investigate the feasibility of self-learning of MLLMs in embodied environments:

- **DG** refers to the results obtained through direct generation from the initial MLLM without any fine-tuning.

- **SC** (Wang et al., 2023a) represents an optimization method of MLLMs through self-consistency. Specifically, we employ multiple chains of thought (CoT) to prompt an MLLM to answer the same question, followed by majority voting. In our experiments, we utilize three different CoTs to guide the MLLM, ultimately voting for the most reasonable action.

- **LMSI** (Huang et al., 2023) is a self-improvement method based on SC. It generates "high-confidence" answers for unlabeled questions to build fine-tuning datasets. This approach enables the LLM to iteratively improve its performance based on the voting mechanism, and we extend this approach to MLLMs.

- **Self-Refine** (Madaan et al., 2023) involves multiple rounds of self-reflection, followed by self-optimization based on the reflection results. This method focuses on prompt optimization and has been validated for feasibility in large-scale LLMs, such as GPT-4. In our experiments, we reflect 3 rounds to get the final result.

- **LLM-Planner** (Song et al., 2023) is a method specifically designed for multimodal embodied planning with text retrieval-augmented techniques. In experiments, we retrieve 3 trajectories as memory for the actor module.

- **RAP** (Kagaya et al., 2024) is a method employing a multimodal retrieval technique for embodied tasks. In our experiments, 3 retrieved trajectories are used for decision-making.

- **Sel-Ask** represents the method that uses grounding guidance prompt engineering techniques from SELU, to substantiate the necessity of critic optimization.

- **SELU-One** represents the method of using the same MLLM to simultaneously perform actor and critic tasks, and fine-tuning with a combination of actor and critic datasets. This approach aims to investigate the feasibility of utilizing a single MLLM to meet the requirements of our framework.

**Evaluation.** For the critic module, we use the MLLM's detection accuracy of whether the task is completed to reflect its ability to understand the environment. Specifically, we provide the last frame of the trajectory during testing and compare whether the MLLM's judgment is consistent with the environmental feedback. For the actor module, the MLLM's decision-making ability is measured by its task success rate. In the experiment, each task is tested 100 times to get the actor MLLM's task success rate. The final frames of these trajectories are then used to evaluate the critic MLLM's success detection accuracy. All results are evaluated at the same dataset. More details for implementation are provided in Appendix A.2.

## 5.2 AI2-THOR

**LLaVA.** We first demonstrate the effectiveness of SELU in the AI2-THOR environment. After online interactions with the environment and fine-tuning of LLaVA, the critic exhibits an average performance improvement of approximately 27%, while the actor achieves an improvement of around 20% compared to the original model. Table 1 and Table 2 present the accuracy of task success detection and task success rate respectively.

Table 1: Accuracy of task success detection in the AI2-THOR environment.

| Method | Pick up | Open | Break | Avg. |
|---|---|---|---|---|
| DG | 80.67% | 36.50% | 50.50% | 55.89% |
| Self-Ask | 79.67% | 41.50% | 53.50% | 58.22% |
| SELU-One | 68.67% | 30.50% | 25.50% | 41.56% |
| SELU | **94.33%** | **67.50%** | **87.50%** | **83.11%** |

Table 2: Task success rate in the AI2-THOR environment. SC and Self-Refine use prompt engineering to realize self-learning, whereas LMSI and SELU utilize fine-tuning. LLM Planner and RAP serve as typical embodied MLLM baselines, incorporating memory retrieval mechanisms.

| Method | Pick Up | Open | Break | Avg. |
|---|---|---|---|---|
| DG | 68.33% | 65.00% | 15.50% | 49.61% |
| SC | 65.67% | 68.50% | 17.50% | 50.56% |
| Self-Refine | 69.67% | 70.50% | 14.50% | 51.56% |
| LMSI | 75.67% | 52.50% | 19.50% | 49.22% |
| LLM-Planner | 66.67% | 52.00% | 11.50% | 43.39% |
| RAP | 62.33% | 63.50% | 14.50% | 46.78% |
| SELU-One | 91.33% | **85.50%** | 27.50% | 68.11% |
| SELU | **94.67%** | 83.50% | **30.50%** | **69.56%** |

In Table 1, it is evidenced that the unified fine-tuning of the actor and the critic (SELU-One) leads to a decline in success detection, even worse than the original critic. Self-Ask proves that fine-tuning is more effective than prompt engineering for the critic module. Although SELU-One achieves a task success rate comparable to that of SELU as shown in Table 2, the compromised critic results in SELU-One incorrectly analyzing the trajectory in subsequent epochs.

Table 2 demonstrates that baselines are not suitable for the self-learning of embodied MLLMs. Both prompt-engineering and fine-tuning baselines struggle to improve the decision-making ability of the MLLM. The reason is that the embodied MLLM cannot give a correct task detection with a lack of environmental understanding. As we can see in Table 1, the initial judgment of the MLLM (DG) on tasks is only about 55%. In this case, merely optimizing the prompt to create multiple CoTs for repeated reflection does not help the MLLM gain task achievement details in an unknown environment. Therefore, neither SC nor Self-Refine can substantially enhance the task success rate. LLM-Planner and RAP also don't increase MLLM's ability to understand the environment. For fine-tuning baselines, relying on statistical voting to validate its own behavior even leads to worse performance. For instance, in the Open task, for LMSI the task detection accuracy of 36.5% causes the actor's performance to drop from 65% to 52.5% after fine-tuning. These results

demonstrate the necessity of the critic module in SELU, and optimizing the critic is crucial for enhancing the actor's performance.

Notably, in the Open task, we can observe a low accuracy of success detection for SELU; however, it still helps the actor improve performance. This highlights the role of hindsight relabeling, which is discussed in detail in Section 5.4.

**Qwen-VL.** In order to prove that SELU can help different MLLMs achieve self-learning, we select Qwen-VL and test it in the AI2-THOR environment under the same setting. The result is shown in Table 3, which indicates that SELU can help Qwen-VL improve the task evaluation capability by about 24% and the decision-making performance by approximately 23%.

Table 3: Self-learning performance of SELU on Qwen-VL in the AI2-THOR environment.

| Task | Qwen-VL-DG | | Qwen-VL-SELU | |
|---|---|---|---|---|
| | Critic | Actor | Critic | Actor |
| Pick Up | 73.33% | 57.67% | **95.67%** | **95.33%** |
| Open | 51.00% | 46.50% | **81.50%** | **68.00%** |
| Break | 63.50% | 12.50% | **83.50%** | **21.50%** |
| Avg. | 62.61% | 38.89% | **86.89%** | **61.61%** |

## 5.3 VirtualHome

We then conduct experiments in the VirtualHome environment, which incorporates a greater variety of items and human agents, thereby enriching the experimental environments to demonstrate the effectiveness of our method. The experimental results are presented in Tables 4 and 5. In this environment, SELU enhances LLaVA task evaluation capability by approximately 30% and improves decision-making performance by around 24%, and it also outperforms baselines. As the environment becomes more complex, the lack of environmental understanding causes SC and Self-Refine to negatively impact the decision-making of the original embodied MLLM. As shown in Table 5, the performance of SC and Self-Refine become even lower than DG.

Table 4: Accuracy of task success detection in the VirtualHome environment.

| Method | Grab | Open | Sit | Avg. |
|---|---|---|---|---|
| DG | 52.67% | 35.33% | 44.50% | 44.17% |
| Self-Ask | 54.33% | 36.67% | 43.50% | 44.83% |
| SELU-One | 45.33% | 15.67% | **48.50%** | 36.50% |
| SELU | **93.67%** | **83.33%** | 47.50% | **74.83%** |

Table 5: Task success rate in the VirtualHome environment.

| Method | Grab | Open | Sit | Avg. |
|---|---|---|---|---|
| DG | 65.00% | 83.33% | 56.50% | 68.28% |
| SC | 52.67% | 81.67% | 61.50% | 65.28% |
| Self-Refine | 59.67% | 74.33% | 60.50% | 64.83% |
| LMSI | 35.67% | 93.67% | 52.50% | 60.61% |
| LLM-Planner | 55.33% | 78.67% | 50.50% | 61.50% |
| RAP | 58.67% | 79.33% | 51.50% | 63.17% |
| SELU-One | 83.67% | **98.67%** | **94.50%** | 92.28% |
| SELU | **93.33%** | 97.67% | 93.50% | **94.83%** |

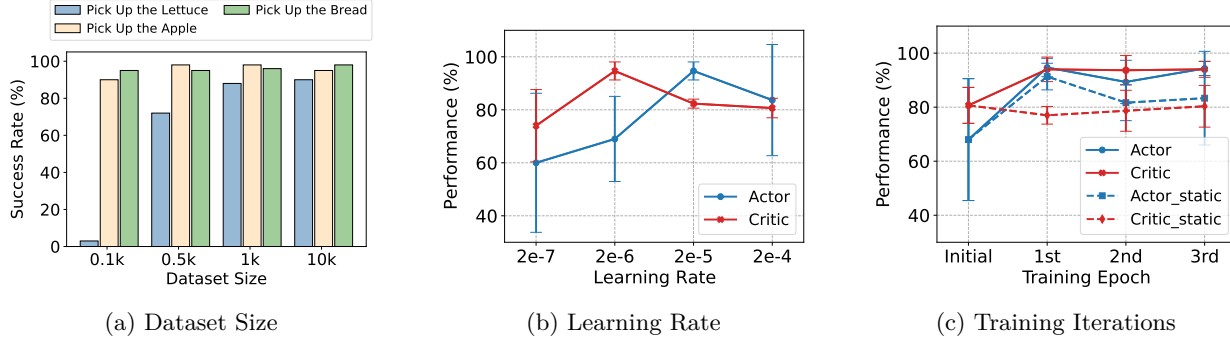

(a) Dataset Size         (b) Learning Rate         (c) Training Iterations

Figure 3: Hyperparameter study of SELU on picking up tasks in the AI2-THOR environment: (a) explores the size of the interaction dataset required for embodied MLLMs, (b) illustrates why a single MLLM is not suitable for SELU from the perspective of learning rare, and (c) demonstrates that the effect of multiple training iterations.

## 5.4 Ablation Study

We conduct ablation experiments on the critic module in the AI2-THOR environemnt, and the results are shown in Table 6. SELU w/o SA means that we do not utilize self-asking to examine the critic output. SELU w/o HR means that we do not perform hindsight relabeling to reanalyze the trajectory when evaluating the task. SELU w/o ALL means that we remove both self-asking and hindsight relabeling and use the data obtained from environment interaction to directly fine-tune the actor. In this case, the evaluation result of the critic is derived from the original MLLM.

Table 6: Ablation study in the AI2-THOR environment.

| Task | Critic (Success Detection Accuracy) | | | | Actor (Task Success Rate) | | | |
| --- | --- | --- | --- | --- | --- | --- | --- | --- |
| | SELU | w/o HR | w/o SA | w/o ALL | SELU | w/o HR | w/o SA | w/o ALL |
| Pick Up | **94.33%** | 83.67% | 78.33% | 80.67% | **94.67%** | 67.33% | 59.67% | 56.33% |
| Open | **67.50%** | 31.50% | 29.00% | 36.67% | **83.50%** | 66.50% | 65.50% | 72.50% |
| Break | **87.50%** | 83.50% | 73.50% | 50.50% | **30.50%** | 27.50% | 23.00% | 17.50% |
| Avg. | **83.11%** | 66.22% | 60.28% | 55.95% | **69.56%** | 57.11% | 49.39% | 48.78% |

By comparing SELU and SELU w/o ALL, we can see the importance of whole critic module clearly. Only by understanding the environment can we achieve the improvement of decision-making in all tasks. By comparing SELU w/o SA and SELU w/o ALL, we find that self-asking can correct the critic's comprehension of the environmental task. It also serves as a premise for hindsight relabeling, as misclassifications without self-asking may lead to relabeling data with an entirely opposite distribution, severely impacting SELU's effectiveness. But reflection on a single task is not enough. We find that the lack of hinsight relabeling leads to a decrease in the diversity of the online fine-tuning dataset, and some tasks can not get enough data to perform well. For example, we can observe a declined performance of success detection and decision-making in Open tasks from SELU w/o ALL to SELU w/o HR. By incorporating hindsight relabeling, from SELU w/o HR to SELU, we can perform a comprehensive multi-task evaluation for each trajectory, ensuring that the embodied MLLM achieves self-learning on each task. Consequently, self-asking and hindsight relabeling are essential components of the critic.

## 5.5 Hyperparameter Analysis

**Online Dataset Size.** Since the MLLMs fine-tuning process is sensitive to the dataset size, we explore the amount of interaction data required to achieve effective learning for embodied tasks. We conduct multiple tests on this variable based on picking up tasks in the AI2-THOR environment. The results are presented in Figure 3a.

The size of dataset is described by the number of trajectories collected for a single task. For instance, dataset-10k indicates that 10k trajectories are collected for a specific task, such as picking up an apple, during online interactions. We evaluate the performance through the actor in terms of task success rate. We can see that the performance of dataset-1k is close to that of dataset-10k. For dataset-10k, it takes about two days to collect one epoch data. To balance experimental performance with sampling efficiency, we opt to sample 1k trajectories per task in our experiments.

Based on this conclusion, we employ a data augmentation method during the actor training process. Since MLLMs tend to prioritize text over images when making decisions, they often focus excessively on the text prompt and overlook image comprehension. To address this issue, we shuffle the action lists in the prompts of training data to provide multiple prompts for the same image. This approach not only increases the dataset size, but also strengthens the connection between the MLLM policy and the observations.

**Learning Rate.** The learning rate is a critical factor, which prevents us from using the same MLLM for the actor and critic. We test different learning rates for the actor and critic separately on picking up tasks in the AI2-THOR, and the results are shown in Figure 3b. We find that, due to the varying sizes of the fine-tuning datasets, using a uniform learning rate inevitably leads to overfitting or underfitting in one of the components, thereby impacting the overall performance of SELU. Therefore, we ultimately use two MLLMs to construct the SELU framework, ensuring effective self-learning. In our experiments, the learning rate for the critic is set to 2e-6 and for the actor is set to 2e-5.

**Training Iterations.** The goal of our framework is to achieve multi-iteration self-learning improvement; therefore, we also evaluate the results of multiple rounds of fine-tuning on picking up tasks in the AI2-THOR environment. To ensure the effectiveness of fine-tuning, we retain 30% of the last fine-tuning dataset each time and obtain the remaining 70% of the data through online interaction. The results are presented in Figure 3c. The solid line represents the performance under an optimized critic, while the dashed line indicates the results under a static critic.

The unstable improvement of the actor under a static critic highlights the necessity of fine-tuning the critic. What's more, our results indicate that multiple iterations of fine-tuning do not consistently improve SELU's performance in every iteration. Both the actor and critic exhibit significant performance improvements during the first iteration of fine-tuning, but show fluctuations and minimal growth in the subsequent iterations. We attribute this limitation to the performance of the critic. The critic performs the success detection based on the last frame, making it difficult to compare the quality of different successful trajectories. Once the actor reaches a level to roughly complete the task, we lack the nuanced supervisory signals to guide the actor for further improvement. Consequently, while there is a notable improvement after the first iteration, subsequent enhancements are limited. We also attempt to increase the number of frames for success detection. However, the performance of the small-scale MLLM does not meet our needs.

# 6 Conclusion and Limitation

In this paper, we introduce SELU, a method for MLLMs to achieve self-learning in unknown environments. SELU facilitates interaction with the environment, analyzes interaction trajectories, builds an online dataset, and performs coupled optimization of the actor and critic. We employ self-asking and hindsight relabeling to enhance the critic task evaluation capabilities. Ablation experiments demonstrate that relabeling significantly expands the critic task judgment range. By leveraging the principle that MLLMs possess stronger perceptual abilities than decision-making abilities, we improve the performance of the actor policy. We test SELU in the AI2-THOR and VirtualHome environments, achieving critic improvements of approximately 28% and 30%, and policy improvements of about 20% and 24%, respectively. Additionally, to validate the applicability of SELU across different MLLMs, we evaluate it on Qwen-VL, resulting in a 23% performance enhancement.

One limitation of SELU is the lack of detailed assessment of successful trajectories. SELU can help embodied MLLM to self-learn how to accomplish tasks in unknown environments. How to complete tasks more efficiently is the next problem to solve. In future work, we will explore finer-grained critic signals to perform more accurate quality assessments of trajectories, guiding embodied MLLMs to tackle more complex tasks, such as long-horizon combination tasks.

As the current simulator masks the agent in the first-person view, we rely on third-person images to assess task completion. For future work, we aim to explore more suitable simulators or deploy SELU in real-world scenarios, enabling a consistent visual perspective for both the actor and critic modules.

## 7 Acknowledgements

This research was supported in part by NSFC under Grants 62173324, 62206281, 62450001 and Doubao Fund.

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

# A  Appendix

## A.1  Pseudocode of SELU

---

**algorithm 1** SELU

---

**Input:** critic MLLM $M_c$, actor MLLM $M_a$, critic fine-tuning dataset $\mathcal{D}_{\text{critic}}$, actor fine-tuning dataset $\mathcal{D}_{\text{actor}}$, maximum timestep $T$, initial instruction list $L$, success detection prompt $p_d$ and action plan prompt $p_a$

**Output:** critic MLLM $M_c$, actor MLLM $M_a$

1: $\mathcal{D}_{\text{critic}}, \mathcal{D}_{\text{actor}} \leftarrow \{\}$
2: **function** "SELU"
3:   **for** instruct $I$ in $L$ **do**
4:    **while** data collecting not done **do**
5:     **for** timestep $t = 1$ to $T$ **do**
6:      get observation $o_t$ from env
7:      $l_{a,t} = M_a(I, p_a, o_t)$
8:      use $l_{a,t}$ to interact with env
9:     **end for**
10:    $l_d = M_c(I, p_d, o_T)$
11:    **if** $l_d =$ "$yes$" **then**
12:     store $(I, p_d, o_T, l_d)$ into $\mathcal{D}_{\text{critic}}$
13:     store $(I, p_a, o_t, l_{a,t}), t = 1, ...T$ into $\mathcal{D}_{\text{actor}}$
14:    **else**
15:     get $l'_d$ through self-asking
16:     **if** $l'_d =$ "$yes$" **then**
17:      store $(I, p_d, o_T, l'_d)$ into $\mathcal{D}_{\text{critic}}$
18:      store $(I, p_a, o_t, l_{a,t}), t = 1, ...T$ into $\mathcal{D}_{\text{actor}}$
19:     **else**
20:      get $I'$ through hindsight relabeling
21:      **if** $I' \neq$ "$None$" **then**
22:       store $(I', p_d, o_T, yes)$ into $\mathcal{D}_{\text{critic}}$
23:       store $(I', p_a, o_t, l_{a,t}), t = 1, ...T$ into $\mathcal{D}_{\text{actor}}$
24:      **end if**
25:     **end if**
26:    **end if**
27:    **end while**
28:   **end for**
29:   optimization $M_c$ and $M_a$ by $\mathcal{D}_{\text{critic}}$ and $\mathcal{D}_{\text{actor}}$
30:   **return** critic MLLM $M_c$, actor MLLM $M_a$
31: **end function**

---

## A.2  Implementation Details

### A.2.1  Environments

Figure 4 shows our experiment environments. Both environments restrict agents to only interact with visible items, limiting their operational range to guarantee behavior plans realistic. Therefore, the actor MLLM makes decisions based on first-person perspective input to ensure accuracy as Figure 4a and Figure 4c show. Given the limitations of the first-person view, the critic MLLM uses a third-person perspective to evaluate the trajectory, reducing hallucinations and obtaining accurate scene information as Figure 4b and Figure 4d show.

The positioning of the third-person camera is crucial, as it should accurately capture the agent's position and the objects it interacts with. Any occlusion or interference can impair the MLLM's understanding of the image, thereby affecting the results of critic success detection and hindsight relabeling.

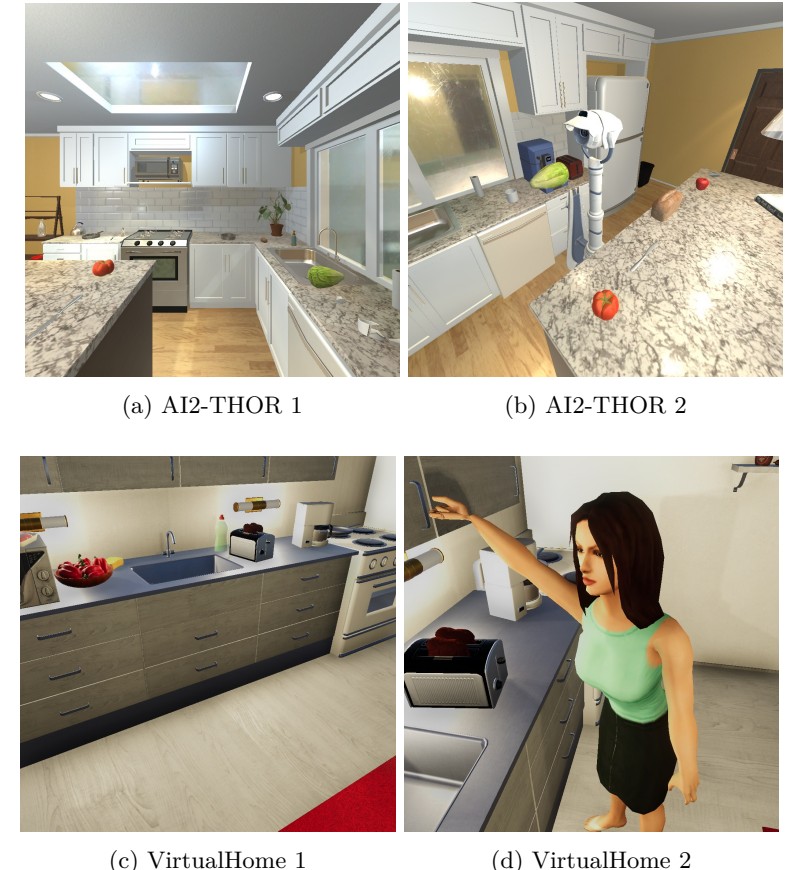

(a) AI2-THOR 1    (b) AI2-THOR 2

(c) VirtualHome 1    (d) VirtualHome 2

Figure 4: The diagram of experimental environments. We utilize the first-person perspective for decision-making and a third-person perspective for trajectory evaluation.

### A.2.2 Hyperparameters

The specific MLLMs we use are LLaVA-V1.6-Mistral-7B and Qwen-VL. We use LoRA to fine-tune them, the hyperparameters are as follows.

Table 7: Hyperparameters of LLaVA Fine-tuning by LoRA.

| Hyperparameters | Value |
|---|---|
| Train_batch_size | 16 |
| Eval_batch_size | 4 |
| Gradient_accumulation_steps | 1 |
| Learning_rate_actor | 2e-5 |
| Learning_rate_critic | 2e-6 |
| Warmup_ratio | 0.03 |
| Weight_decay | 0.0 |
| Model_max_length | 2048 |
| Lr_scheduler_type | cosine |
| Tf32 | True |

Table 8: Hyperparameters of Qwen-VL Fine-tuning by LoRA.

| Hyperparameters | Value |
|---|---|
| Train_batch_size | 2 |
| Eval_batch_size | 1 |
| Gradient_accumulation_steps | 8 |
| Learning_rate_actor | 1e-5 |
| Learning_rate_critic | 1e-6 |
| Warmup_ratio | 0.01 |
| Weight_decay | 0.1 |
| Adam_beta2 | 0.95 |
| Model_max_length | 2048 |
| Lr_scheduler_type | cosine |
| Bf16 | True |
| Lazy_preprocess | True |

Both models are configured with a temperature of 0 and a maximum token length of 2048 for response generation. The maximum number of environment steps is set to 10. The multiple chains of thought for self-consistency (Wang et al., 2023a) , and the multiple rounds of reflection in self-refine (Madaan et al., 2023) are set to 3, following the settings reported in their papers. For LLM-Planner (Song et al., 2023) and RAP (Kagaya et al., 2024), retrieval parameters follow the official implementation to ensure consistency. The number of retrieved trajectories is fixed at the minimum value of 3, as adopted in RAP. The list of hyperparameters is provided in Table 9.

Table 9: Hyperparameters List for Baselines.

| Hyperparameters | Value |
|---|---|
| Response generation temperature | 0 |
| Maximum token length | 2048 |
| Maximum environment steps | 10 |
| Number of CoTs (Wang et al., 2023a) | 3 |
| Number of reflection rounds (Madaan et al., 2023) | 3 |
| Retrieve buffer | 20 |
| KNN retrieves (LLM Planner (Song et al., 2023)) | 9 |
| Multimodal retrieves (RAP (Kagaya et al., 2024)) | 5 |
| Retrieve trajectories input | 3 |

### A.2.3 Low-Level Actions of Agents

SELU is designed to enable embodied agents to learn atomic skills through MLLMs themselves. In our experiments, we use the "locobot" agent in AI2-THOR and the "female1" agent in VirtualHome. The set of low-level actions to these agents is summarized in Table 10.

Table 10: Low-Level Actions of AI2-THOR and VirtualHome.

| Action | Env | Description |
|---|---|---|
| **MoveAhead**, **MoveBack** | AI2-THOR/VirtualHome | Move the agent forward or backward by a fixed step (0.25 meters). |
| **RotateLeft**, **RotateRight** | AI2-THOR/VirtualHome | Rotate the agent 90° to the left or right. |
| **Pick** | AI2-THOR/VirtualHome | Pick up a visible and graspable object. The action fails if the object is not visible. |
| **Open** | AI2-THOR/VirtualHome | Open an object that is both interactable and openable. |
| **Break** | AI2-THOR | Break an object that is both interactable and breakable. |
| **Sit** | VirtualHome | Make the agent sit on an object that is both interactable and sittable. |

### A.3 Prompts for SELU

The specific prompts we use for SELU are:

---

**Actor-Interaction with Env**

This is the current observation from a {agent} in a {AI2-THOR/VirtualHome} environment. Now the {agent} needs to finish the task {instruction}, you can only choose the following action to interact with the environment, which are {action_list}. If you choose {PickupObject/GrabObject}, OpenObject, {BreakObject/SitObject}, you should give a specific object name. Now the objects you can interact with are {visible_objs_str}. What's your next action to implement the command to {instruction}? You should output your action and the reasoning. The output format should be:
Action:...
Object:...
Reasoning:...

---

**Critic-Success Detection**

The image shows a third-person view from the {agent}'s perspective in a {AI2-THOR/VirtualHome} environment. Please check whether the {instruction.objects} in the image is {instruction.verb → adj.} or not? You should output yes or no, and the reasoning. The output format should be:
Result:...
Reasoning:...

---

**Critic-Self Asking 1**

The image shows a third-person view from the {agent}'s perspective in a {AI2-THOR/VirtualHome} environment. Please check the state of the {instruction.objects} in the image. You should output the state and the reasoning. The output format should be:
State:...
Reasoning:...

---

---

**Critic-Self Asking 2**

The image shows a third-person view from the {agent}'s perspective in a {AI2-THOR/VirtualHome} environment. The {instruction.objects} in the observation is in {objects.state} state, please determine whether the {instruction} has been completed or not. You should output yes or no, and the reasoning. The output format should be:
Result:...
Reasoning:...

---

**Critic-hindsight relabeling 1**

The image shows a third-person view from the {agent}'s perspective in a {AI2-THOR/VirtualHome} environment. Please see the image carefully. Determine whether there is any object that is {instruction.verb → adj.} by the {agent}? You should output the object name and the reasoning. The output format should be:
Object:...
Reasoning:...

---

**Critic-hindsight relabeling 2**

The image shows a third-person view from the {agent}'s perspective in a {AI2-THOR/VirtualHome} environment. The {relabeling.object} in the observation is {instruction.verb → adj.}, you should give a new instruction based on it. The original instruction is {instruction}, what's the new instruction? The output format should be:
New instruction:...
Reasoning:...

---

### A.4 Visualization of Actor and Critic on LLaVA in AI2-THOR

The embodied actor MLLM is used to interact with the unknown environment, and collect trajectories from the evaluation of the critic MLLM. An example for 'pick up the lettuce' is shown in figure 5.

The embodied critic MLLM is used to perform success detection on each trajectory and use self-asking and hindsight relabeling techniques to build bootstrapped dataset. An example for 'break the mug' is shown in figure 6.

### A.5 Multi-Frame Critic Study

To further justify our choice of using the final frame, we conduct the frame number parameter experiments to see its effect. As LLaVA can only input a single image, we use half of the pretraining dataset to fine-tune it first to allow multi-image input. The critic performance is evaluated using the DG baseline setting, with detection accuracy of task completion. The results are shown in Table 11. We conduct the parameter experiments with three configurations: the final frame (DG-1), the last 3 frames (DG-3), and the last 5 frames (DG-5).

Table 11: Multi-frame Evaluation for Critic Module.

|        | Pick Up    | Open       | Break      | Avg.       |
| ------ | ---------- | ---------- | ---------- | ---------- |
| DG-1   | **80.67%** | 36.50%     | 50.50%     | **55.89%** |
| DG-3   | 76.67%     | **37.50%** | 51.00%     | 55.06%     |
| DG-5   | 73.67%     | **37.50%** | **53.50%** | 54.89%     |

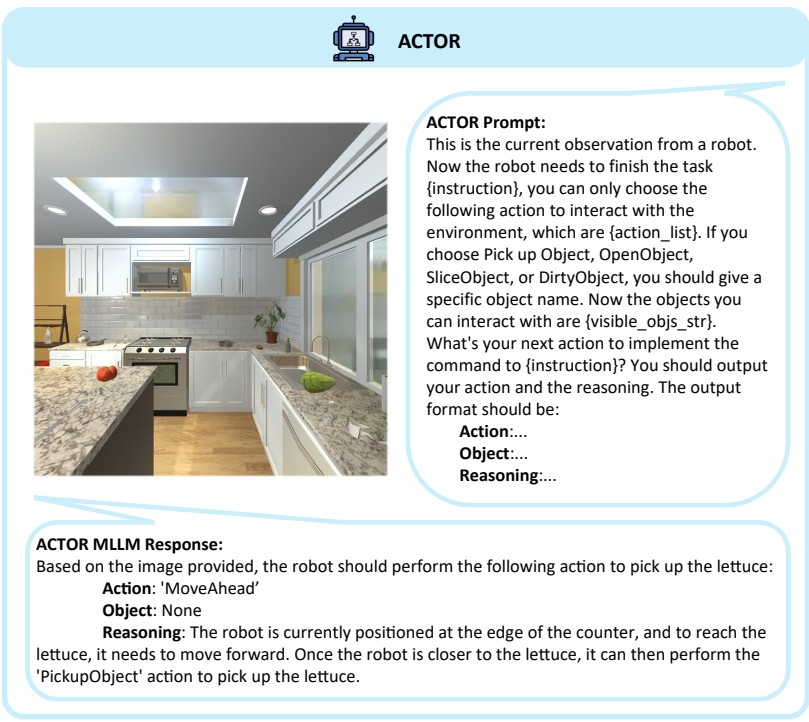

Figure 5: A visualization of the actor MLLM interacting with the AI2-THOR environment. The agent is instructed to pick up the lettuce. As the lettuce is far away, the agent needs to move closer before attempting to pick it up.

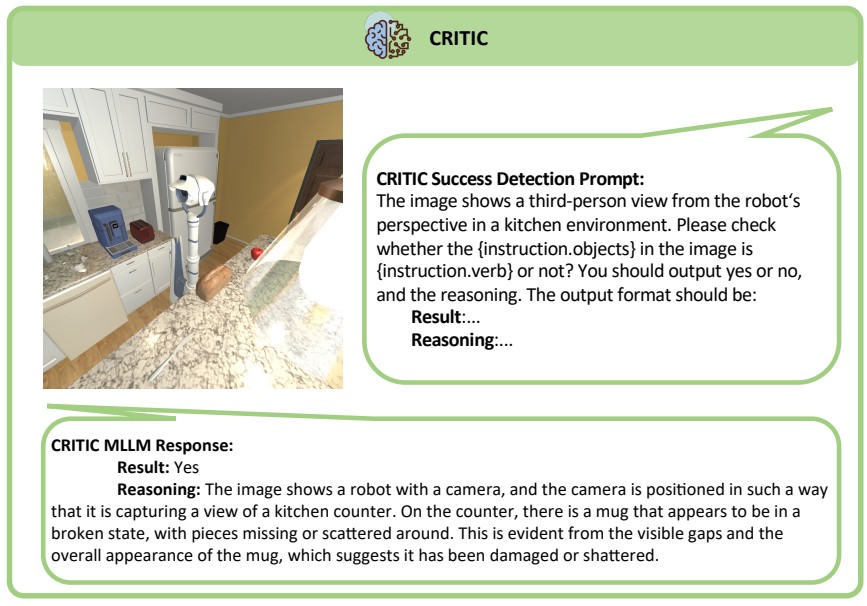

Figure 6: A visualization of the critic MLLM conducting success detection on a trajectory. Since the trajectory has completed the 'break the mug' task, it can be added to the fine-tuning dataset.

The evaluation with multiple images does not outperform single-image evaluation. There are two reasons to explain this result. First, most embodied simulators employ flash-like transitions (Li et al., 2025) to represent interaction processes. This shortcoming stems from the fact that different agents often require

distinct low-level control policies. To ensure reliable and generalizable execution, simulators always use flash-like transitions for the interaction process. Consequently, multi-images are not helpful to understand the task completion. Second, current MLLMs still exhibit limitations in spatial reasoning (Chen et al., 2024a). Whether provided with egocentric or third-person views, MLLMs can capture the task-related objects but can not give a clear description of the position relationship between the robot and these objects. Given these constraints, we adopt the final frame as the input for the task critic.

## A.6 Computational Resource Costs

We run all experiments in 8 x A100 GPUs with 40GB memory. The details of computational resource costs can be seen in the Table 12. The training time and GPU hours reflect the training cost, while peak VRAM and inference time represent the inference cost. While SELU incurs higher training costs, its inference speed is quicker than traditional baselines as they need more time for retrieval techniques.

Table 12: Computational Resource Costs of LLaVA.

| Method | Training Time (h) | GPU Hours (h) | Peak VRAM (GB) | Inference Time (s/sample) |
| --- | --- | --- | --- | --- |
| DG | None | None | 22.57 | 1.26 |
| SC | None | None | 22.62 | 4.13 |
| LMSI | 5.97 | 47.76 | 22.57 | 1.31 |
| Self-Refine | None | None | 22.57 | 5.22 |
| LLM-Planner | None | None | **23.21** | 2.76 |
| RAP | None | None | **23.21** | **5.71** |
| Self-Ask | None | None | 22.57 | 2.56 |
| SELU-One | 6.65 | 53.20 | 22.57 | 4.72 (critic) 1.26 (actor) |
| SELU | **6.72** | **53.76** | 22.57 | 4.72 (critic) 1.26 (actor) |

## A.7 Name-Removed Prompt Experiments

As introduced in Section 1, an unknown environment refers to one in which the MLLM has neither been exposed to its visual data during pretraining, nor fine-tuned on its task-specific data to achieve grounding. The environment name does not significantly impact the MLLM's initial perception and decision-making ability. This is demonstrated by the experimental results in Table 13 and Table 14, where we replace "a {AI2-THOR/VirtualHome} environment" in our prompts with "a household environment". As the MLLM is unknown to the environment, the name does not notably affect its performance.

Table 13: Critic Performance of LLaVA in AI2-THOR Environment with Name-Removed Prompt

| Method | Pick Up | Open | Break | Avg. |
| --- | --- | --- | --- | --- |
| DG | 78.33% | 38.50% | 51.50% | 56.11% |
| Self-Ask | 78.67% | 43.00% | 56.50% | 59.39% |
| SELU-One | 65.67% | 33.50% | 31.00% | 43.39% |
| SELU | **91.67%** | **67.00%** | **89.50%** | **82.72%** |

## A.8 Long-Horizon Task Experiments

In this work, the self-learning MLLM is designed to learn atomic skills to finish short-horizon tasks. As embodied MLLMs have been proved to have the ability to decompose complex and long-horizon tasks into atomic skills. Several benchmarks, such as ALFRED, have been developed to evaluate this planning ability. If we can design a self-learning method to help the MLLM to autonomously acquire atomic skills, it will

Table 14: Actor Performance of LLaVA in AI2-THOR Environment with Name-Removed Prompt

| Method | Pick Up | Open | Break | Avg. |
|---|---|---|---|---|
| DG | 66.67% | 66.50% | 13.50% | 48.89% |
| SC | 67.33% | 65.00% | 14.50% | 48.94% |
| Self-Refine | 72.33% | 69.50% | 15.50% | 52.44% |
| LMSI | 77.67% | 60.50% | 11.50% | 49.89% |
| LLM-Planner | 69.67% | 53.50% | 13.50% | 45.56% |
| RAP | 66.67% | 67.50% | 14.50% | 49.56% |
| SELU-One | 89.33% | 80.50% | 29.50% | 66.44% |
| SELU | **93.67%** | **83.00%** | **31.00%** | **69.22%** |

significantly facilitate the collection of diverse data and improve the MLLM's capabilities. In ALFRED, long-horizon tasks typically span around 50 timesteps and can be decomposed into 5-6 subtasks by an MLLM planner. Based on this, we set a horizon of 10 timesteps for the MLLM to learn each atomic skill.

To further evaluate the scalability of SELU, we extend the horizon length to 40 timesteps and use the same exploring prompt to initialize the task list. The objects are increased to 8-10 (e.g., 10 for "pick up" and "break" tasks, and 8 for "open" tasks). The corresponding experimental results are shown in Table 15 and Table 16. SELU still performs well in learning long-horizon atomic skills. However, in practice, we often decompose these skills into shorter segments to reduce the complexity of low-level skill execution for embodied robots.

Table 15: Critic Performance of LLaVA in AI2-THOR Environment with 40 Timesteps

| Method | Pick Up | Open | Break | Avg. |
|---|---|---|---|---|
| DG | 73.30% | 31.13% | 49.30% | 51.24% |
| Self-Ask | 74.20% | 33.63% | 51.60% | 53.14% |
| SELU-One | 51.70% | 26.50% | 36.20% | 38.13% |
| SELU | **89.50%** | **69.38%** | **89.10%** | **82.66%** |

Table 16: Actor Performance of LLaVA in AI2-THOR Environment with 40 Timesteps

| Method | Pick Up | Open | Break | Avg. |
|---|---|---|---|---|
| DG | 42.80% | 51.38% | 12.70% | 35.63% |
| SC | 45.30% | 51.75% | 12.10% | 36.38% |
| Self-Refine | 47.20% | 55.13% | 15.30% | 39.21% |
| LMSI | 51.60% | 59.38% | 13.70% | 41.56% |
| LLM-Planner | 44.70% | 56.50% | 16.30% | 39.17% |
| RAP | 45.60% | 55.25% | 17.90% | 39.58% |
| SELU-One | 81.30% | **86.75%** | 31.40% | 66.48% |
| SELU | **83.50%** | 84.38% | **32.70%** | **66.86%** |

