# OpenReview forum: "SELU: Self-Learning Embodied Multimodal Large Language Models in Unknown Environments"
_TMLR — Accepted by TMLR_

### Review · Reviewer_72Xe · 2025-06-07

**Summary Of Contributions:**

The paper proposes SELU, an actor–critic style loop for embodied MLLMs in unknown environments. A critic MLLM refines its own success-detection ability through “self-asking” (focus questions) and “hindsight relabeling” (converting failed trajectories into success samples for other tasks); an actor MLLM is then fine-tuned on trajectories the critic judges as successful. Coupled optimisation is shown on AI2-THOR and VirtualHome with LLaVA and Qwen-VL back-bones, achieving SOTA results.

**Audience:**

Yes

**Broader Impact Concerns:**

No.

**Claims And Evidence:**

Yes

**Requested Changes:**

1. Explain why the critic is fed only the final frame and report an ablation that varies the number or range of input frames (e.g., last-1, last-3, full trajectory). This will clarify whether “last-frame” is a design choice or a limitation.
2. Add training time, GPU hours, peak VRAM, and inference speed for SELU and all baselines so readers can gauge practical cost.
3. Report performance when each component is removed individually and when both are removed, to isolate their true contributions (self-asking and hindsight relabeling).

**Strengths And Weaknesses:**

Strengths:
1. The problem is interesting to make the MLLM as the learnable critic without external feedback.
2. The paper is easy to follow.
3. SELU achieves competitive results on 2 benchmarks.

Weakness
1. Critic looks only at the last frame and cannot compare trajectory quality. When the critic judges success from a single image, it may mislabel longer actions.
2. Each epoch still needs around 1 k trajectories per task and collecting 10k trajectories for one task costs two days on AI2-THOR, which seems to be less efficient. Moreover, since SELU has separate actor and critic, which doubles GPU memory and training time, please also provide the information about GPU memory, training time and inference time compared with baselines.
3. Ablation studies are incomplete. It is better to separately analyze the function of self-asking and hindsight relabeling. Moreover, more qualitative examples should be shown in illustrate how hindsight relabeling corrects the trajectory label.

---

> ### Author Response · Authors · 2025-07-17
> **Rebuttal with Reviewer 72Xe**
>
> We sincerely appreciate the reviewers for their valuable feedback. We hope our following answers will clear up the doubts about our work, and please let us know if there is any other clarification we can provide.
>
> ---
>
> **Q1: Explain why the critic is fed only the final frame and report an ablation.**
>
> **A1:**  There are two main reasons for our choice of using the final frame for the critic module.
>
> First, embodied simulators typically use flash-like transitions[1] due to varying low-level policies across agents, making multi-images less informative for task evaluation.
>
> Second, current MLLMs still exhibit limitations in spatial reasoning [2]; they can recognize task-relevant objects but fail to accurately describe their spatial relationships. Thus, MLLMs are not suitable for tracking task progress.
>
> Given these constraints, we adopt the final frame as the input for the task critic. We add the ablation experiment as suggested in Appendix A.5 to further justify our choice of using the final frame. As shown in the table below, multi-frame evaluation does not outperform single-frame evaluation.
> **Table10: Multi-frame Evaluation for Critic Module**
> |       | **Pick Up** | **Open**  | **Break** | **Average** |
> |-------|-------------|-----------|-----------|-------------|
> | **DG-1**  | **80.67%**  | 36.50%    | 50.50%    | **55.89%**  |
> | **DG-3**  | 76.67%      | **37.50%**| 51.00%    | 55.06%      |
> | **DG-5**  | 73.67%      | **37.50%**| **53.50%**| 54.89%      |
> [1]Kolve, Eric, et al. "Ai2-thor: An interactive 3d environment for visual ai." _arXiv preprint arXiv:1712.05474_ (2017).
> [2]Chen B, Xu Z, Kirmani S, et al. Spatialvlm: Endowing vision-language models with spatial reasoning capabilities[C]//Proceedings of the IEEE/CVF Conference on Computer Vision and Pattern Recognition. 2024: 14455-14465.
>
> ---
> **Q2: Add computing resources cost for SELU and all baselines.**
>
> **A2:**  Thanks for your valuable suggestions to make our experiments details clearer. We add the computing resources cost in our Appendix A.6. We run all experiments in 8 x A100 GPUs with 40GB memory.
> **Table11: Computing resources cost of LLaVA in AI2-THOR experiments**
> |                            | **DG** | **SC** | **LMSI** | **Self-Refine** | **LLM-Planner** | **RAP** | **Self-Ask** | **SELU-One**                 | **SELU**                      |
> |----------------------------|--------|--------|----------|------------------|------------------|--------|-------------|------------------------------|-------------------------------|
> | **Training Time (h)**      | None   | None   | 5.97     | None             | None             | None   | None        | 6.65                         | **6.72**                          |
> | **GPU Hours (h)**          | None   | None   | 47.76    | None             | None             | None   | None        | 53.2                         | **53.76**                         |
> | **Peak VRAM (GB)**         | 22.57  | 22.62  | 22.57    | 22.57            | **23.21**            | **23.21**  | 22.57       | 22.57                        | 22.57                         |
> | **Inference Time (s/sample)** | 1.26   | 4.13   | 1.31     | 5.22             | 2.76             | **5.71**   | 2.56        | 4.72 (critic)  1.26 (actor)                | 4.72 (critic)  1.26 (actor)               |
> ---
> **Q3: Report performance when each component is removed individually.**
>
> **A3:** Thank you for your valuable suggestions. We update the ablation results to better highlight the individual contributions of the self-asking and hindsight relabeling components. The reason we do not directly apply the hindsight relabeling module is that it requires the MLLM to use self-asking first. Without this step, misclassifications may cause the relabeling process to generate data with an entirely opposite distribution, severely impacting SELU's effectiveness. This corresponding analysis is also added to the ablation experiment analysis.
> **Table 6: Ablation study in the AI2-THOR environment.**
> | Task    | Critic          |    (Success     |    Detection    |     Accuracy)    | Actor            |    (Task     |    Success     |     Rate)    |
> |---------|---------------------------------------------|--------|--------|---------|--------------------------------------|--------|--------|---------|
> |         |  **SELU**  | **w/o HR** | **w/o SA** | **w/o ALL** | **SELU**   | **w/o HR** | **w/o SA** | **w/o ALL** |
> | **Pick Up** | **94.33%** | 83.67% | 78.33% | 80.67%  | **94.67%** | 67.33% | 59.67% | 56.33%  |
> | **Open**    | **67.50%** | 31.50% | 29.00% | 36.67%  | **83.50%** | 66.50% | 65.50% | 72.50%  |
> | **Break**   | **87.50%** | 83.50% | 73.50% | 50.50%  | **30.50%** | 27.50% | 23.00% | 17.50%  |
> | **Avg.**| **83.11%** | 66.22% | 60.28% | 55.95% | **69.56%** | 57.11% | 49.39% | 48.78% |

---

> > ### Author Response · Authors · 2025-07-17
> > **Ignore This Comment**
> >
> > Please ignore this comment due to formatting issues.

---

> ### Author Response · Authors · 2025-07-17
> **Rebuttal with Reviewer 72Xe**
>
> We sincerely appreciate the reviewers for their valuable feedback. We hope our following answers will clear up the doubts about our work, and please let us know if there is any other clarification we can provide.
>
> ---
>
> **Q1: Explain why the critic is fed only the final frame and report an ablation.**
>
> **A1:**  There are two main reasons for our choice of using the final frame for the critic module.
>
> First, embodied simulators typically use flash-like transitions[1] due to varying low-level policies across agents, making multi-images less informative for task evaluation.
>
> Second, current MLLMs still exhibit limitations in spatial reasoning [2]; they can recognize task-relevant objects but fail to accurately describe their spatial relationships. Thus, MLLMs are not suitable for tracking task progress.
>
> Given these constraints, we adopt the final frame as the input for the task critic. We add the ablation experiment as suggested in Appendix A.5 to further justify our choice of using the final frame. As shown in the table below, multi-frame evaluation does not outperform single-frame evaluation.
> **Table10: Multi-frame Evaluation for Critic Module**
> |       | **Pick Up** | **Open**  | **Break** | **Average** |
> |-------|-------------|-----------|-----------|-------------|
> | **DG-1**  | **80.67%**  | 36.50%    | 50.50%    | **55.89%**  |
> | **DG-3**  | 76.67%      | **37.50%**| 51.00%    | 55.06%      |
> | **DG-5**  | 73.67%      | **37.50%**| **53.50%**| 54.89%      |
> [1]Kolve, Eric, et al. "Ai2-thor: An interactive 3d environment for visual ai." _arXiv preprint arXiv:1712.05474_ (2017).
> [2]Chen B, Xu Z, Kirmani S, et al. Spatialvlm: Endowing vision-language models with spatial reasoning capabilities[C]//Proceedings of the IEEE/CVF Conference on Computer Vision and Pattern Recognition. 2024: 14455-14465.
>
> ---
> **Q2: Add computing resources cost for SELU and all baselines.**
>
> **A2:**  Thanks for your valuable suggestions to make our experiments details clearer. We add the computing resources cost in our Appendix A.6. We run all experiments in 8 x A100 GPUs with 40GB memory.
>
> **Table11: Computing resources cost of LLaVA in AI2-THOR experiments**
>
> |                            | **DG** | **SC** | **LMSI** | **Self-Refine** | **LLM-Planner** | **RAP** | **Self-Ask** | **SELU-One**                 | **SELU**                      |
> |----------------------------|--------|--------|----------|------------------|------------------|--------|-------------|------------------------------|-------------------------------|
> | **Training Time (h)**      | None   | None   | 5.97     | None             | None             | None   | None        | 6.65                         | **6.72**                          |
> | **GPU Hours (h)**          | None   | None   | 47.76    | None             | None             | None   | None        | 53.2                         | **53.76**                         |
> | **Peak VRAM (GB)**         | 22.57  | 22.62  | 22.57    | 22.57            | **23.21**            | **23.21**  | 22.57       | 22.57                        | 22.57                         |
> | **Inference Time (s/sample)** | 1.26   | 4.13   | 1.31     | 5.22             | 2.76             | **5.71**   | 2.56        | 4.72 (critic)  1.26 (actor)                | 4.72 (critic)  1.26 (actor)               |
> ---
> **Q3: Report performance when each component is removed individually.**
>
> **A3:**  Thank you for your valuable suggestions. We update the ablation results to better highlight the individual contributions of the self-asking and hindsight relabeling components. The reason we do not directly apply the hindsight relabeling module is that it requires the MLLM to use self-asking first. Without this step, misclassifications may cause the relabeling process to generate data with an entirely opposite distribution, severely impacting SELU's effectiveness. This corresponding analysis is also added to the ablation experiment analysis.
>
> **Table 6: Ablation study in the AI2-THOR environment.**
>
> | Task    | Critic          |    (Success     |    Detection    |     Accuracy)    | Actor            |    (Task     |    Success     |     Rate)    |
> |---------|---------------------------------------------|--------|--------|---------|--------------------------------------|--------|--------|---------|
> |         |  **SELU**  | **w/o HR** | **w/o SA** | **w/o ALL** | **SELU**   | **w/o HR** | **w/o SA** | **w/o ALL** |
> | **Pick Up** | **94.33%** | 83.67% | 78.33% | 80.67%  | **94.67%** | 67.33% | 59.67% | 56.33%  |
> | **Open**    | **67.50%** | 31.50% | 29.00% | 36.67%  | **83.50%** | 66.50% | 65.50% | 72.50%  |
> | **Break**   | **87.50%** | 83.50% | 73.50% | 50.50%  | **30.50%** | 27.50% | 23.00% | 17.50%  |
> | **Avg.**| **83.11%** | 66.22% | 60.28% | 55.95% | **69.56%** | 57.11% | 49.39% | 48.78% |

---

> ### Author Response · Authors · 2025-07-17
> **Rebuttal with Reviewer 72Xe**
>
> We sincerely appreciate the reviewers for their valuable feedback. We hope our following answers will clear up the doubts about our work, and please let us know if there is any other clarification we can provide.
>
> ---
>
> **Q1: Explain why the critic is fed only the final frame and report an ablation.**
>
> **A1:**  There are two main reasons for our choice of using the final frame for the critic module.
>
> First, embodied simulators typically use flash-like transitions[1] due to varying low-level policies across agents, making multi-images less informative for task evaluation.
>
> Second, current MLLMs still exhibit limitations in spatial reasoning [2]; they can recognize task-relevant objects but fail to accurately describe their spatial relationships. Thus, MLLMs are not suitable for tracking task progress.
>
> Given these constraints, we adopt the final frame as the input for the task critic. We add the ablation experiment as suggested in Appendix A.5 to further justify our choice of using the final frame. As shown in the table below, multi-frame evaluation does not outperform single-frame evaluation.
>
> **Table10: Multi-frame Evaluation for Critic Module**
>
> |       | **Pick Up** | **Open**  | **Break** | **Average** |
> |-------|-------------|-----------|-----------|-------------|
> | **DG-1**  | **80.67%**  | 36.50%    | 50.50%    | **55.89%**  |
> | **DG-3**  | 76.67%      | **37.50%**| 51.00%    | 55.06%      |
> | **DG-5**  | 73.67%      | **37.50%**| **53.50%**| 54.89%      |
>
> [1]Kolve, Eric, et al. "Ai2-thor: An interactive 3d environment for visual ai." _arXiv preprint arXiv:1712.05474_ (2017).
>
> [2]Chen B, Xu Z, Kirmani S, et al. Spatialvlm: Endowing vision-language models with spatial reasoning capabilities[C]//Proceedings of the IEEE/CVF Conference on Computer Vision and Pattern Recognition. 2024: 14455-14465.
>
> ---
> **Q2: Add computing resources cost for SELU and all baselines.**
>
> **A2:**  Thanks for your valuable suggestions to make our experiments details clearer. We add the computing resources cost in our Appendix A.6. We run all experiments in 8 x A100 GPUs with 40GB memory.
>
> **Table11: Computing resources cost of LLaVA in AI2-THOR experiments**
>
> |                            | **DG** | **SC** | **LMSI** | **Self-Refine** | **LLM-Planner** | **RAP** | **Self-Ask** | **SELU-One**                 | **SELU**                      |
> |----------------------------|--------|--------|----------|------------------|------------------|--------|-------------|------------------------------|-------------------------------|
> | **Training Time (h)**      | None   | None   | 5.97     | None             | None             | None   | None        | 6.65                         | **6.72**                          |
> | **GPU Hours (h)**          | None   | None   | 47.76    | None             | None             | None   | None        | 53.2                         | **53.76**                         |
> | **Peak VRAM (GB)**         | 22.57  | 22.62  | 22.57    | 22.57            | **23.21**            | **23.21**  | 22.57       | 22.57                        | 22.57                         |
> | **Inference Time (s/sample)** | 1.26   | 4.13   | 1.31     | 5.22             | 2.76             | **5.71**   | 2.56        | 4.72 (critic)  1.26 (actor)                | 4.72 (critic)  1.26 (actor)               |
> ---
> **Q3: Report performance when each component is removed individually.**
>
> **A3:**  Thank you for your valuable suggestions. We update the ablation results to better highlight the individual contributions of the self-asking and hindsight relabeling components. The reason we do not directly apply the hindsight relabeling module is that it requires the MLLM to use self-asking first. Without this step, misclassifications may cause the relabeling process to generate data with an entirely opposite distribution, severely impacting SELU's effectiveness. This corresponding analysis is also added to the ablation experiment analysis.
>
> **Table 6: Ablation study in the AI2-THOR environment.**
>
> | Task    | Critic          |    (Success     |    Detection    |     Accuracy)    | Actor            |    (Task     |    Success     |     Rate)    |
> |---------|---------------------------------------------|--------|--------|---------|--------------------------------------|--------|--------|---------|
> |         |  **SELU**  | **w/o HR** | **w/o SA** | **w/o ALL** | **SELU**   | **w/o HR** | **w/o SA** | **w/o ALL** |
> | **Pick Up** | **94.33%** | 83.67% | 78.33% | 80.67%  | **94.67%** | 67.33% | 59.67% | 56.33%  |
> | **Open**    | **67.50%** | 31.50% | 29.00% | 36.67%  | **83.50%** | 66.50% | 65.50% | 72.50%  |
> | **Break**   | **87.50%** | 83.50% | 73.50% | 50.50%  | **30.50%** | 27.50% | 23.00% | 17.50%  |
> | **Avg.**| **83.11%** | 66.22% | 60.28% | 55.95% | **69.56%** | 57.11% | 49.39% | 48.78% |
> ---

---

> ### Author Response · Authors · 2025-07-22
> **Rebuttal with Reviewer 72Xe**
>
> Thank you for your valuable feedback. We have uploaded the latest revision of the paper. We hope that the revised **Ablation Study,  Appendix A.5 and Appendix A.6 Section** of the paper can resolve your doubts, and we would be grateful for any further comments or suggestions you may have.

---

> > ### Comment · Reviewer_72Xe · 2025-08-03
> > **Comments**
> >
> > Thanks for the detailed rebuttal from the author. My concerns listed  are well addressed. Please combine the rebuttal into the final paper.

---

> > > ### Author Response · Authors · 2025-08-03
> > > **Rebuttal with Reviewer 72Xe**
> > >
> > > Thank you for acknowledging the newly added experiments and revised sections. The updated version now incorporates all the details discussed, and we will ensure that these are preserved in the final paper.

---

### Review · Reviewer_UatL · 2025-06-16

**Summary Of Contributions:**

This paper proposes a new method SElf-Learning paradigm in Unknown environments (SELU), which aims to train multi-model large language models (MLLMs) to perform actions in a simulated environment. To achieve this, SELU uses an actor-critic method where the critic MLLM judges whether the actor LLM achieved the goals given to the actor. If the critic finds the actor has not achieved its goal, the MLLM relables the trajectory to synthetically generate a successful rollout.

**Audience:**

Yes

**Broader Impact Concerns:**

I see no broader impact concerns

**Claims And Evidence:**

No

**Requested Changes:**

# Would strengthen work

Remove the somewhat arbitrary distinctions between SELU and “normal” RL like
> With the development of MLLMs, the feedback provided to the agent is no longer constrained to scalar values, like rewards; it can now include diverse modalities, such as natural language (Dong et al., 2024).

Ultimately, you still produce scalar (in fact binary) rewards, they are just not explicitly written down.

Run a “prompt SELU” variant to check whether the Actor-Critic filtering or the model training causes the uplift in performance.

Compare performance between “reason-then-act” and the current “act-then-reason” prompt-designs.

Discuss the impact of estimation errors and compounding errors on Agent/Critic training.


# Necessary for acceptance

Reframe the „unknown environments“ claim or supply more experiments in more environments. Generally, I recommend removing the information of whether the model sees AI2-THOR or VirtualHome from the prompt to preempt contamination of the pretraining dataset (for unknown environments you wouldn’t be able to name the environment anyways). This also goes to dependent statements like
> Compared to existing studies, we focus on the self-learning potential of MLLMs, as external feedback may be not professional enough in unknown environments.

This is not supported by your experiments since they are not “unknown environments” (both in the narrow sense that these are well known simulators, but also in the wide sense that the real-world spaces are well known already).

In the main text or appendix, list every primitive action available to the agent.

Remove the list of visible objects from the prompt. When studying MLLMs Vision should be enough.

Increase task-horizon lengths (e.g., 40–50 steps) and object/task variety to test scalability.

Describe verb‐extraction algorithms in pseudocode. Also describe more thoroughly whether the relabeling is “open book vs “closed book”.

Use a consistent viewpoint (either always first-person or always third-person) for both actor and critic and report the results. The system should perform well in both cases.

**Strengths And Weaknesses:**

# Strengths

The hindsight replay in application to robotics is novel and interesting. The closest thing is probably https://arxiv.org/pdf/2302.05206 which also does hindsight relabeling, though not in a robotics context (though this shouldn’t matter for RL methods).

The writing is clear and the method is generally well described.

Using hindsight relabeling is a good idea to use the already existing knowledge inside pretrained MLLMs.

# Weaknesses

The claim of an “Unknown” environment paradigm (SELU) is misleading: the agent is explicitly provided the environment space (e.g., AI2-THOR/VirtualHome). Beyond that, real-world environments cannot be truly unknown. This is why most methods working with LLMs+RL work on e.g. code since it’s easier to estimate whether the model can truly perform in arbitrary environments (though I do understand that as soon as you include vision this is no longer possible).

The distinction between RL and SELU is really arbitrary: SELU is RL with binary returns (and relabeling). Consider an agent with {0,1} returns, then a naïve policy gradient would compute $-\log(\pi(a|s))\cdot1$ for the 1 rewards  and $-\log(\pi(a|s))\cdot0=0$ for the failures. By only training your agent on the successful rollouts, you end up doing exactly this.
Keeping this in mind, one should also compare against Motif https://arxiv.org/pdf/2310.00166 which uses preference pairs to build the reward instead of a binary returns as in SELU.

The experiments are way too limited in complexity: Task horizons are capped at 10 steps with only 2–3 objects and 3 task types, which is too small for generalization. More complex tasks with 4–5× longer horizons are needed, given that low-level control is abstracted away. It's not clear why this method wouldn't collapse as the task horizon is increased.

Many of your baselines also do not train a model, so comparing against SELU is not entirely fair, considering that SELU can use up to 10k examples just for training. You could try to run SELU with the dataset as examples in your prompts to validate whether the actor-critic component gives the performance uplift or just the fact that you train the models.

Critics’ misdetections, even if rare, compound over self-bootstrapped training data, reinforcing model biases. This is a known issue in synthetic/intrinsic-reward RL that’s not discussed. Even if you have a 90% success chance, the missing 10% might introduce a compounding bias during training.
This becomes especially relevant if you are in novel environments since this increases the likelihood of misdetections.

There are some missing aspects for the method implementation as well:

- Action Space: No full description of possible actions.
- Verb Extraction & Relabeling Mechanics: It’s unclear how verbs are extracted, what aspects of trajectories may be relabeled, and whether relabeling has access to which nouns or words are possible (“open book vs “closed book” relabeling).

Looking at the prompt template: Reasoning is placed after action selection; intuitively, prompting the model to “think” before choosing an action (so it can reflect on its arguments) would be more natural and potentially more effective.

Why do you list the objects the model can interact with?
> “Now the objects you can interact with are {visible_objs_str}.”

Shouldn’t this be identifiable from the vision itself? Otherwise, why do you need the vision component?

There is also a major limitation in how information is presented to the MLLM: Figure 4 say that a first-person view is used for acting but a third-person view for critiquing. Why? Especially in the real world, having two completely independent camera systems is not realistic. This could also explain why SELU-One does not perform well. If SELU works in unknown environments, it should also be robust against changes in POV. I recommend you run the experiments with both models using first person and both third person.

Minor note: The acronym SELU conflicts with “Scaled Exponential Linear Units” in existing ML literature.

---

> ### Author Response · Authors · 2025-07-20
> **Rebuttal with Reviewer UatL (1/2)**
>
> We sincerely appreciate the reviewers for their valuable feedback. We hope our following answers will clear up the doubts about our work, and please let us know if there is any other clarification we can provide.
>
> ---
>
> **Q1: Reframe the "unknown environments" claim and supply experiments with name-removed prompt.**
>
> **A1:** Thank you for your valuable suggestions. In response, we add further clarification regarding the definition of the **unknown environment** in our paper, and we conduct an additional experiment—**removing the environment name from the prompt**—as you suggested.
>
> Our work focuses on enabling self-learning in MLLMs for embodied tasks. An **unknown environment** refers to one in which the MLLM has neither been **fine-tuned** with data from that environment to achieve grounding nor encountered its visual content during **pretraining**. For example, the pretraining data[1] of LLaVA does not include images from the AI2-THOR and VirtualHome environments. It is also not fine-tuned with their online data to achieve grounding. Therefore, we can claim that these two embodied environments are **unknown** to LLaVA. While the model may acquire general perception and decision-making capabilities from other household-related datasets, our DG experiment results indicate that its performance in these unknown environments is still limited. To make the definition of an unknown environment clearer to readers, we add a detailed explanation in the third paragraph of the **Introduction**.
>
> Additionally, as suggested, we conduct the requested experiment in **Appendix A.7**. In this experiment, we **remove** the environment's name from the prompt. The experimental results show that the performance remains consistent with or without the environment name. As MLLM is unknown to the environment, the name does not significantly affect its perception and decision-making abilities.
>
> [1] LLaVA pretrain dataset: https://github.com/haotian-liu/LLaVA/blob/main/docs/Data.md
>
> **Table 11: Critic Performance of LLaVA in AI2THOR Environment with Name-Removed Prompt**
> | Method     | Pick Up | Open   | Break  | Avg.   |
> |------------|---------|--------|--------|--------|
> | DG         | 78.33%  | 38.50% | 51.50% | 56.11% |
> | Self-Ask   | 78.67%  | 43.00% | 56.50% | 59.39% |
> | SELU-One   | 65.67%  | 33.50% | 31.00% | 43.39% |
> | SELU       | **91.67%**  | **67.00%** | **89.50%** | **82.72%** |
>
> **Table 12: Actor Performance of LLaVA in AI2THOR Environment with Name-Removed Prompt**
> | Method       | Pick Up | Open   | Break  | Avg.   |
> |--------------|---------|--------|--------|--------|
> | DG           | 66.67%  | 66.50% | 13.50% | 48.89% |
> | SC           | 67.33%  | 65.00% | 14.50% | 48.94% |
> | Self-Refine  | 72.33%  | 69.50% | 15.50% | 52.44% |
> | LMSI         | 77.67%  | 60.50% | 11.50% | 49.89% |
> | LLM-Planner  | 69.67%  | 53.50% | 13.50% | 45.56% |
> | RAP          | 66.67%  | 67.50% | 14.50% | 49.56% |
> | SELU-One     | 89.33%  | 80.50% | 29.50% | 66.44% |
> | SELU         | **93.67%**  | **83.00%** | **31.00%** | **69.22%** |
>
> ---
>
> **Q2: self-learning baseline choice and "external feedback may be not professional enough in unknown environments" conclusion.**
>
> **A2:** In this paper, we explore **self-learning** approaches for **embodied MLLMs**. Existing baselines in this field can be broadly categorized into two types: **self-learning by prompt engineering** (e.g., SC, Self-Refine) and **self-learning by fine-tuning** (e.g., LMSI). To provide a more comprehensive evaluation, we also include traditional **embodied MLLM planning methods**, such as LLM-Planner and RAP.
>
> The concern that **"external feedback may not be professional enough"** is raised by Tianlu Wang et al. [2], who argue that even human-labeled datasets used for LLM judgment training can be **noisy** and often require MLLMs to identify and select **high-quality annotations**. This conclusion highlights the motivation behind the development of self-learning methods. And we put this sentence in the **Related Work Section** to emphasize the importance of self-learning.
>
> [2] Wang T, Kulikov I, Golovneva O, et al. Self-taught evaluators[J]. arXiv preprint arXiv:2408.02666, 2024.
>
> ---
>
> **Q3: List every primitive action available to the agent.**
>
> **A3:** Thank you for your helpful suggestions. To clarify the experimental details, we add the action space table in **Appendix A.2.3**.
>
> ---
>
> **Q4: Remove the list of visible objects from the prompt. When studying MLLMs Vision should be enough.**
>
> **A4:** This is a hindsight relabeling technique. For small-scale models, hindsight relabeling cannot be achieved very well. The list of visible objects is not directly obtained from the environment but generated by the **MLLM itself**. Like a Chain-of-Thought (CoT) prompt, this approach helps clarify the interactive objects, thereby increasing the likelihood of correct hindsight relabeling.
>
> ---

---

> ### Author Response · Authors · 2025-07-20
> **Rebuttal with Reviewer UatL (2/2)**
>
> **Q5: Increase task-horizon lengths (e.g., 40–50 steps) and object/task variety to test scalability.**
>
> **A5:** Thank you for your concern regarding **task complexity**. We want to take this opportunity to clarify the **motivation** behind SELU and to introduce the **common setting** of embodied tasks.
>
> In this work, the self-learning MLLM is designed to learn **atomic skills** to finish short-horizon tasks. As embodied MLLMs have been proved to have the ability to **decompose** complex and long-horizon tasks into atomic skills. Several benchmarks—such as ALFRED[3]—have been developed to evaluate this **planning ability**. If we can design a self-learning method to help the MLLM to **autonomously** acquire atomic skills, it will significantly facilitate the collection of diverse data and improve the MLLM's capabilities. In ALFRED, long-horizon tasks typically span around **50 timesteps** and can be decomposed into **5–6 subtasks** by an MLLM planner. Based on this, we set a horizon of **10 timesteps** for the MLLM to learn each atomic skill.
>
> To further evaluate the scalability of SELU, we extend the horizon length to **40 timesteps** and use the same exploring prompt to initialize the task list. The objects are increased to **8-10** (e.g., 10 for "pick up" and "break" tasks, and 8 for "open" tasks). The corresponding experimental results are added in **Appendix A.8**. As shown, SELU still performs well in learning long-horizon atomic skills. However, in practice, we often decompose these skills into shorter segments to reduce the complexity of low-level skill execution for embodied robots.
>
> [3] Shridhar M, Thomason J, Gordon D, et al. Alfred: A benchmark for interpreting grounded instructions for everyday tasks[C]//Proceedings of the IEEE/CVF conference on computer vision and pattern recognition. 2020: 10740-10749.
>
> **Table 13: Critic Performance of LLaVA in AI2THOR Environment with 40 Timesteps**
> | Method     | Pick Up | Open   | Break  | Avg.   |
> |------------|---------|--------|--------|--------|
> | DG         | 73.30%  | 31.13% | 49.30% | 51.24% |
> | Self-Ask   | 74.20%  | 33.63% | 51.60% | 53.14% |
> | SELU-One   | 51.70%  | 26.50% | 36.20% | 38.13% |
> | SELU       | **89.50%**  | **69.38%** | **89.10%** | **82.66%** |
>
> **Table 14: Actor Performance of LLaVA in AI2THOR Environment with 40 Timesteps**
> | Method       | Pick Up | Open   | Break  | Avg.   |
> |--------------|---------|--------|--------|--------|
> | DG           | 42.80%  | 51.38% | 12.70% | 35.63% |
> | SC           | 45.30%  | 51.75% | 12.10% | 36.38% |
> | Self-Refine  | 47.20%  | 55.13% | 15.30% | 39.21% |
> | LMSI         | 51.60%  | 59.38% | 13.70% | 41.56% |
> | LLM-Planner  | 44.70%  | 56.50% | 16.30% | 39.17% |
> | RAP          | 45.60%  | 55.25% | 17.90% | 39.58% |
> | SELU-One     | 81.30%  | **86.75%** | 31.40% | 66.48% |
> | SELU         | **83.50%**  | 84.38% | **32.70%** | **66.86%** |
>
> ---
>
> **Q6: Describe verb‐extraction algorithms and relabeling process.**
>
> **A6:** Thank you for your valuable suggestions to improve the clarity of our paper. We will include detailed descriptions of both components in the **Method section**. The verb-extraction algorithm employs a classic **format-matching** approach, which requires the MLLM to strictly follow the prompt format and extract the verb text by identifying specific fragments. In the relabeling process, an atomic skill primarily consists of two elements: **actions** and **interactive objects**. The actions are **"closed-book"**, meaning they are restricted to the actions that the robot can perform. The objects, on the other hand, are **"open-book"**, as the MLLM identifies visible objects by itself and selects from them.
>
> ---
>
> **Q7: Use a consistent viewpoint for actor and critic.**
>
> **A7:** Thank you for your suggestion regarding the use of a consistent viewpoint. We adopt a combination of first-person and third-person views for **two main reasons**. First of all, we would like to clarify that using different viewpoints is **common** in embodied AI scenarios[4], particularly when addressing both interaction and evaluation tasks. In real-world tasks, combining global and local cameras helps mitigate **occlusion issues**. Especially, in embodied simulators as we discussed with Reviewer 72Xe in Q1, the flash-like transitions make it more challenging to assess task completion due to frequent occlusions. Second, the simulator **masks** the robot in the first-person view, which significantly impairs the analysis of its interactions. Therefore, we employ multiple viewpoints to ensure more accurate and comprehensive evaluation.
>
> [4] Chen Y, Tian S, Liu S, et al. Conrft: A reinforced fine-tuning method for vla models via consistency policy[J]. arXiv preprint arXiv:2502.05450, 2025.
>
> ---

---

> ### Author Response · Authors · 2025-07-22
> **Rebuttal with Reviewer UatL**
>
> Thank you for your valuable feedback. We have uploaded the latest revision of the paper. We hope that the **revised Method, Appendix A.2.3, Appendix A.7 and Appendix A.8 Section** of the paper can resolve your doubts, and we would be grateful for any further comments or suggestions you may have.

---

> > ### Comment · Reviewer_UatL · 2025-07-28
> >
> > Thank you for providing the additional results
> >
> > > Reframe the "unknown environments" claim and supply experiments with name-removed prompt.
> >
> > The results seem stable from the case without the environment name, so this should clear up any data-leakage concerns
> >
> > > List every primitive action available to the agent.
> >
> > Ok, the list of action seems reasonable (though pretty high level)
> >
> > > Remove the list of visible objects from the prompt. When studying MLLMs Vision should be enough. The list of visible objects is not directly obtained from the environment but generated by the MLLM itself. Like a Chain-of-Thought (CoT) prompt, this approach helps clarify the interactive objects, thereby increasing the likelihood of correct hindsight relabeling.
> >
> > Ah, if this is part of the MLLM detection, then this is fine
> >
> > > Increase task-horizon lengths (e.g., 40–50 steps) and object/task variety to test scalability. To further evaluate the scalability of SELU, we extend the horizon length to 40 timesteps and use the same exploring prompt to initialize the task list
> >
> > This clears that concern
> >
> > > Use a consistent viewpoint for actor and critic
> >
> > I still think this is a limitation: Yes, prior work also uses 3d person view, but this is due to them using external feedback. If an independent observe "sees" the action from outside and rates it, a third person view is the correct viewpoint to use, but here, where an agent is supposed to identify completion in an unsupervised manner, the same viewpoint should be usable (especially if the goal is to have an embodied agent). Problems like occlusion or general partial observability are challenges you will run into in this self-labeling setting. However, I also understand that the simulation might simply not provide enough data due to flash transitions to make first person analysis viable. How about a compromise: Just note this as a potential for future work.
> >
> > Otherwise all my concerns have been cleared.

---

> > > ### Author Response · Authors · 2025-07-28
> > > **Rebuttal with Reviewer UatL**
> > >
> > > Thank you for acknowledging the newly added experiments and revised sections.
> > >
> > > Regarding the limitation of **a consistent viewpoint**, we agree that it is a valid concern. This compromise was made after we attempted to use the same perspective, which unfortunately led to suboptimal experimental results. We think your suggestion is highly valuable, and we have added a corresponding discussion in the future work section **(Section 6)** to improve the overall rigor of the paper.
> > >
> > > The latest revision of the paper has been uploaded.

---

### Review · Reviewer_6R2S · 2025-07-14

**Summary Of Contributions:**

This paper proposes an actor-critic self-learning framework, named SELU, designed to enhance the decision-making capabilities of multimodal large language models (MLLMs). In this framework, the critic leverages self-asking and hindsight relabeling techniques to extract knowledge from interaction trajectories generated by the actor, thereby improving its understanding of the environment. In turn, the actor benefits from self-feedback provided by the critic, leading to more informed decision-making. The effectiveness of SELU is demonstrated in the AI2-THOR and VirtualHome environments, where it yields improvements of approximately 28% and 30% for the critic, and 20% and 24% for the actor through self-learning. While actor-critic frameworks have previously been employed to enhance decision-making in LLMs, the key contribution of this work lies in successfully extending this paradigm to multimodal LLMs.

**Audience:**

Yes

**Claims And Evidence:**

Yes

**Requested Changes:**

n/a

**Strengths And Weaknesses:**

Strengths:
While actor-critic frameworks have been utilized to improve decision-making in LLMs, the primary contribution of this work lies in effectively adapting and applying this paradigm to multimodal large language models (MLLMs).

Weaknesses:
As noted above, actor-critic frameworks have already been shown to be effective in enhancing the decision-making capabilities of LLMs. Therefore, it is not particularly surprising that a similar effect is observed in MLLMs. From an algorithmic perspective, the paper offers limited novelty. Techniques such as self-asking and hindsight relabeling are not new and have been explored in prior work, see. e.g.,

@article{gaven2024sac,
  title={SAC-GLAM: Improving Online RL for LLM agents with Soft Actor-Critic and Hindsight Relabeling},
  author={Gaven, Loris and Romac, Clement and Carta, Thomas and Lamprier, Sylvain and Sigaud, Olivier and Oudeyer, Pierre-Yves},
  journal={arXiv preprint arXiv:2410.12481},
  year={2024}
}

---

> ### Author Response · Authors · 2025-07-17
> **Rebuttal with Reviewer 6R2S**
>
> We sincerely appreciate the reviewers for their valuable feedback. We hope our following answers will clear up the doubts about our work, and please let us know if there is any other clarification we can provide.
>
> ---
> **Q1: The paper offers limited novelty compared with prior work.**
>
> **A1:**  Thank you for your thoughtful feedback on the novelty aspect of SELU. We understand your concerns and would like to clarify the distinct motivation and setting of our work.
>
> Our proposed method, SELU, is specifically designed to address the challenge of **self-learning** for MLLMs. Unlike prior approaches that typically rely on human-labeled data or pre-defined reward signals, SELU enables MLLMs to autonomously acquire **new atomic skills** in unknown embodied environments without **external feedback**. This key distinction allows SELU to reduce human involvement and generate a broader and more diverse data distribution for training embodied MLLMs.
>
> The Playground-text environment used in **SAC-GLAM** provides **pre-defined rewards**. We discuss these methods in Figure 1(a) of our paper. As these signals are human-involved and not always available in the real world, we aim to develop a self-learning method that can improve the performance of MLLMs by themselves.
>
> As SAC-GLAM is a concurrent work, it was not included in our initial submission. We add a detailed discussion and comparison of this work in the Related Work section to better clarify distinctions between our approach and others.
>
> ---

---

> > ### Author Response · Authors · 2025-07-22
> > **Rebuttal with Reviewer 6R2S**
> >
> > Thank you for your valuable feedback. We have uploaded the latest revision of the paper. We hope that the **revised Introduction section** of the paper can resolve your doubts, and we would be grateful for any further comments or suggestions you may have.

---

### Decision · Action_Editor_aig6 · 2025-08-13

**Recommendation:** Accept as is

**Additional Comments:**

I recommend acceptance of this paper. SELU presents a novel and well-justified actor–critic self-learning framework that enables MLLMs to self-improve in the absence of external feedback, with strong empirical gains across two established embodied environments. The authors have addressed reviewer concerns comprehensively in their rebuttal.

For the final version, they should incorporate the requested clarifications and results, ensuring clarity on ablations, trajectory selection, and implementation specifics as discussed during the rebuttal phase.

**Audience:**

Yes

**Audience Explanation:**

The topic, i.e., self-learning embodied MLLMs in unknown environments, addresses growing interest in autonomous learning, embodied AI, and multi-modal reasoning. Researchers working on RL, embodied agents, self-supervised learning, and LM-based decision systems would find the paper relevant.

**Claims And Evidence:**

Yes

**Claims Explanation:**

SELU is an actor–critic-inspired self-learning paradigm designed to enable multimodal large language models (MLLMs) to autonomously improve both environmental comprehension and decision-making in environments under unavailability of external feedback. The critic uses self-asking and hindsight relabeling to learn from trajectories obtained using the actor, augmenting its understanding of the unknown environment. The actor updates its policy using feedback from the critic. The experiments show an improvement of >28% for the critic and >20% for the actor, only achieved through self-supervised learning.

The paper presents clear, empirically validated claims, i.e., that SELU enables multimodal LLMs to self-improve within unknown environments. The claims are supported by experiments in simulated embodied environments (AI2-THOR and VirtualHome). The methodology is explained with sufficient detail to ensure reproducibility and clarity.